

# Information flow in non-unitary quantum cellular automata

Elisabeth Wagner[1*], Ramil Nigmatullin[1,2], Alexei Gilchrist[1] and Gavin K. Brennen[1]

**1** Centre for Engineered Quantum Systems, Dept. of Physics & Astronomy,
Macquarie University, 2109 NSW, Australia
**2** Quantinuum, 13-15 Hills Road, CB2 1NL Cambridge, United Kingdom

* elisabeth.wagner@students.mq.edu.au

## Abstract

The information flow in a quantum system is a fundamental feature of its dynamics. An important class of dynamics are quantum cellular automata (QCA), systems with discrete updates invariant in time and space, for which an index theory has been proposed for the quantification of the net flow of quantum information across a boundary. While the index is rigid in the sense of begin invariant under finite-depth local circuits, it is not defined when the system is coupled to an environment, i.e. for non-unitary time evolution of open quantum systems. We propose a new measure of information flow for non-unitary QCA denoted the information current which is not rigid, but can be computed locally based on the matrix-product operator representation of the map.



# 1   Introduction

The essential physical principles of causality and conservation of information impose strong constraints on the time evolution of physical systems. In particular, in the simplest setting where space and time are discrete and causality is preserved, quantum many-body systems can be described by quantum cellular automata (QCA) [1–3], which are systems with discrete variables evolving under a local update rule (in analogy with classical cellular automata). Despite these seemingly crude approximations for realistic many-body dynamics, QCA provide useful models to study different aspects of non-equilibrium physics. Local quantum circuits, a subclass of QCA, have recently received significant attention in connection to questions related to quantum chaos and information scrambling [4–15].

In the past decade, a great deal of progress has been made in the comprehensive characterization of QCA. A classification using index theory was first introduced in [16] for one-dimensional systems and has recently been extended and generalized to higher dimensional systems in [17–19]. In one dimension, the index (or GNVW index according to the acronym of the authors) describes the net flow of quantum information along a chain of, say qubits; while in higher dimensions it is given by the information flow between two subsystems of the total quantum grid of logical qubits. For unitary one-dimensional systems it takes the form of a positive rational fraction, ind $\in \mathbb{Q}_+$, which can be interpreted as the ratio of the number of orthonormal states transferred to the right divided by the number transferred to the left after each discrete time step. Besides its fundamental interest, the index theory has turned out to have practical implications, allowing, for instance, for a classification of 2D Floquet phases exhibiting bulk many-body localization (MBL) [2, 20–26], where the index serves as a topological invariant that measures the chirality of quantum information flow. Formally, the GNVW index has been defined originally in terms of abstract observable algebras [16] which were later argued to be "lacking an immediate physical interpretation" [27] and to be "both physically opaque and not amenable to experimental measurement" [26].

Subsequently, an equivalent definition of the index has been found by taking the entanglement of the "vectorized" evolution operator, or operator-space entanglement entropy, into

consideration. The Rényi-$\alpha$ entropy has been shown to be an appropriate alternative measure, as it can be computed locally and closely reflects the intuitive interpretation of the index in terms of quantum information flow. Using this quantity, any sub-linear entanglement-growth behavior in nontrivial QCA could be ruled out, and a lower bound on quantum chaos has been defined for any Rényi-$\alpha$ entropy of the evolution operator [27].

The original GNVW index has been rediscovered by taking the Rényi-2 entropy into account. This has the advantage that it is a quantity that can be measured directly using "SWAP"-based many-body quantum interference measurements, realisable e.g. using hard-core bosonic ultracold atoms in a shaken optical lattice [20]. The formulation of the index in terms of the Rényi-2 entropy is notably equivalent to a previous derivation of the index in terms of the chiral mutual information [26], as the latter can be constructed from any extensive entanglement measure, including Rényi entropies.

Additionally, matrix product unitaries (MPUs) have turned out to provide a natural framework for the index theory as they have been shown to in fact be QCA and vice versa; i.e. MPUs feature a causal cone, strictly propagating information over a finite distance only. MPUs are thereby guaranteed to preserve locality by mapping local operators to local operators while at the same time all locality-preserving unitaries can be represented in a matrix product way. The index theory implies that all locality-preserving 1D unitaries can be efficiently simulated by MPUs, and that different MPU representations of the same unitary can be related through a local gauge. The explicit computability of the GNVW index via MPUs has been demonstrated in [20], and has led to further physical consequences in the framework of Floquet dynamics, where bulk topology has been shown to enforce chaotic dynamics at the edge.

An equivalent expression of the GNVW index has been given by the "rank-ratio" index, which is defined as the ratio between the ranks of the left and right singular value decompositions of the tensor representing the MPU [28, 29]. Based on this definition, an index theorem for generalized MPUs has been defined taking fermionic QCA into account, where a graded canonical form has been introduced for fermionic matrix product states [30].

Further, Hamiltonian evolutions on the lattice satisfying Lieb-Robinson bounds, rather than strict locality, have been described by approximately locality preserving unitaries (ALPUs). The index theory has been shown to be robust to this generalization, and has been extended to one-dimensional ALPUs classifying a wider class of natural systems with approximate causal cones only. A converse to the Lieb-Robinson bounds has further been achieved, where any ALPU of index zero can be exactly generated by some time-dependent, quasi-local Hamiltonian in constant time. For the special case of finite chains with open boundaries, any unitary satisfying the Lieb-Robinson bound may be generated by such a Hamiltonian [31].

While the aforementioned results provide a deep understanding of information flow in unitary QCA, counterpart measures for discrete non-unitary systems representing more general (irreversible) physical actions [32–34] are missing. In this work, we address this gap and present a measure for the net information current for non-unitary QCA – a class of dynamics which is described by one-dimensional matrix product operators (MPOs) and which can be computed by examining local dynamics only.

To provide the reader with a first intuitive understanding of how the information flow in a quantum system can be defined, we motivate the discussion with a simple example, the reset-swap map, which is elaborated Sec. 3.2. Referring to Fig. 1, the pink vertical line in the center represents the boundary across which the information flow shall be measured. The QCA acts on two qubits at a time, updating nearest neighbor pairs of lattice sites at locations $\{2j, 2j + 1\}_{j \in \mathbb{Z}}$, followed by the same update on pairs shifted by one lattice site. The composition of the two updates together constitutes a single time step of the QCA. Each two-qubit update consists of three local operations: first, a quantum channel, indicated by a box, which resets the qubit on the left hand site of the pair to the state $|0\rangle$; second, an identity opera-

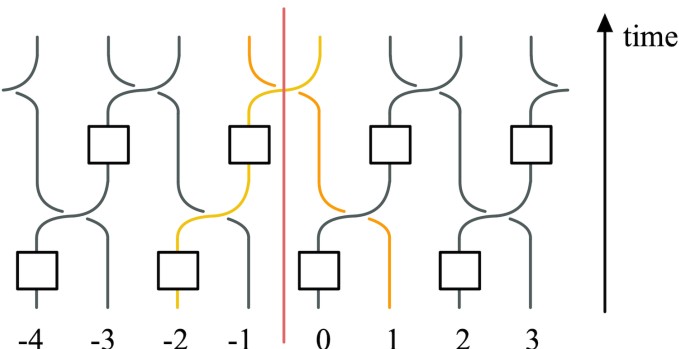

Figure 1: Illustration of one step of a local non-unitary QCA, the reset-swap map described in Sec. 3.2, consisting of local reset onto the state $|0\rangle\langle 0|$ on every second site (indicated by boxes) followed by a pair-wise swap operation. Since only the $|0\rangle\langle 0|$ operator is propagated to the right while all single site operators are propagated to the left there is a net flow of information to the left across the boundary (pink vertical line) in distinction to local unitary QCA where there is none.

tion on the right cell, indicated by a straight line; and third, a swap operation, represented by crossed lines, which swaps the locations of the neighboring qubits. One can see that only the yellow and orange colored worldlines of the initial operators at sites -2 and 1 cross the boundary after one QCA step. Operators on the yellow path are reset by the map and thus only the operator $\{|0\rangle\langle 0|\}$ is transported to the right, while on the orange path all four orthonormal operators $\{|0\rangle\langle 0|, |0\rangle\langle 1|, |1\rangle\langle 0|, |1\rangle\langle 1|\}$ are transported to the left. Thus, in distinction to the unitary case in which no information flow is present for a local unitary rule, a net flow of quantum information occurs to the left for this non-unitary local QCA.

In Sec. 2, we propose such a measure, the information current, and compare and contrast it to the corresponding index for unitary QCA. The physics of the current is illustrated for a variety of physically motivated examples in Sec. 3. Finally, we conclude with a summary of results and open questions in Sec. 4.

## 2 Quantifying information flow

In the following, the mathematical background of the MPO description of QCA is presented in Sec. 2.1, before providing a short summary of the on MPUs based index theory for unitary QCAs in Sec. 2.2. Sec. 2.3 outlines the definition of the information current for non-unitary QCA, whose properties are listed and discussed in the final Sec. 2.4.

### 2.1 MPO description of QCA

In this framework, a single time step of the QCA is modeled by an MPO in the most general form, see Fig. 2(a), with the same local tensor $M$ of the MPO distributed equally across the lattice. These tensors represent superoperators acting on vectorized density matrices in a doubled Hilbert space $\mathcal{H} \times \mathcal{H}^*$.

The MPO in Fig. 2(a) represents the general form of the dynamical map. It can be represented by the circuit shown in Fig. 2(b) when the QCA is exclusively defined by *local* operation. This class of QCA will be referred to as "local QCA" throughout this work. In the referred circuit, local operators $V$ (framed in yellow) act on pairs of neighboring sites, after which the set of operators $W$ (marked with red) update the next pairs of neighboring sites, shifted by one lattice site. This is the simplest, and most commonly used partitioning scheme of QCA with a

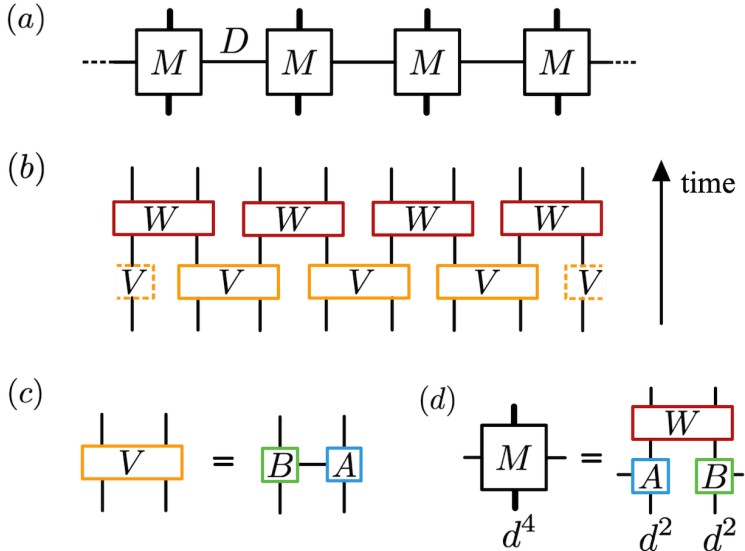

Figure 2: (a) One time step of a QCA, written as an MPO with virtual bond dimension $D$. It acts on a doubled Hilbert space $\mathcal{H} \times \mathcal{H}^*$, and is in (b), (c), and (d) assumed to be locally generated. Its action on two qudits is represented as a vectorization of operators with physical dimension $(d^2)^2 = d^4$. (b) Partitioning scheme of the MPO in (a) for local QCA (excluding e.g. the shift map). (c) Singular value decomposition of the tensor operator $V$. Note that the horizontal bond dimension $D$ is equal to the rank of the operator singular value decomposition of $V$. (d) Definition of the constituent local tensor $M$ of the MPO.

two-cell neighborhood. Note that generality is provided nonetheless, as the sites of QCA with larger interaction neighborhoods can be grouped together, such that it has the same structure as a QCA with a two-cell neighborhood (analogous to a coarse-graining process). Further, the singular value decomposition (SVD) is applied by rewriting one of the operators that acts on two neighboring sites, e.g. $V$, into a single index sum of tensor products of operators $B$ and $A$ which act on the associated left or right site, respectively, according to Fig. 2(c). Then the local tensors $M$ in Fig. 2(a) can be defined according to Fig. 2(d) – as constituent four-index tensors, whose (vertical) physical indices have been grouped together.

On the basis of the MPU description of QCA, an index theory has been formulated for unitary QCA; it is summarized below including a reformulation of its definition.

## 2.2 Index theory for unitary QCA using MPUs

Following Refs. [28, 29], we define the matrices $M_L$ and $M_R$ with input and output Hilbert spaces as indicated in Fig. 3. In [29] it is shown that for unitary one-dimensional QCA, one can quantify the net flow of quantum information to the right via the so-called rank-ratio index:[1]

$$\begin{aligned}
\text{ind} &= \frac{1}{2}(\log \text{Rank}(M_R) - \log \text{Rank}(M_L)) \\
&= \frac{1}{2} \log \left( \frac{\text{Rank}(M_R)}{\text{Rank}(M_L)} \right).
\end{aligned} \tag{1}$$

---

[1]Throughout we take logarithms base 2.

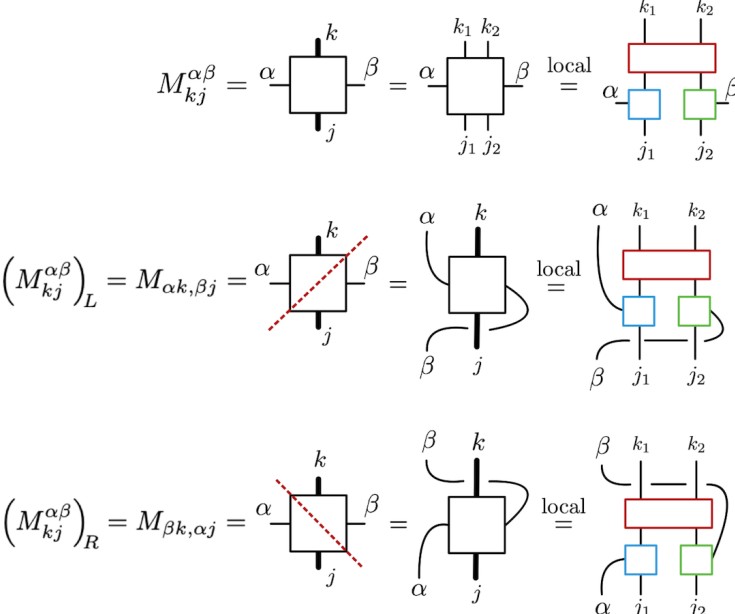

Figure 3: Diagrammatic description of the matrix components of $M_L$ and $M_R$ obtained by taking left and right partitionings of the local tensor $M$ of the MPO that describes the QCA. $\alpha$ and $\beta$ are *virtual* indices, while $j = (j_1, j_2)$ and $k = (k_1, k_2)$ represent the composed *physical* indices of the input or output state, respectively. The adjoint matrix components $\left( M_{kj}^{\alpha\beta} \right)^{\dagger}_{L,R}$ are obtained by reflecting the diagrams about the horizontal axis and replacing the constituent tensors by their adjoints. Note the equations on the furthest right site are only true if the QCA is local.

In anticipation of our alternative measure for information flow below, we note that because Rank$(A) = $ Rank$(A^{\dagger}A)$ for any complex matrix $A$, the index can also be written as

$$\text{ind} = \frac{1}{2}(S_0(\sigma_R) - S_0(\sigma_L)), \tag{2}$$

where

$$\sigma_{\beta} = M_{\beta}^{\dagger} M_{\beta} / \text{Tr}\left[ M_{\beta}^{\dagger} M_{\beta} \right], \quad \text{for } \beta \in \{L, R\}, \tag{3}$$

are trace one, positive, Hermitian operators. Here $S_0(\rho) = \log \text{Rank}(\rho)$, also known as the Hartley entropy, is the $\alpha = 0$ case of the Rényi-$\alpha$ entropy

$$S_{\alpha}(\rho) = \frac{1}{1-\alpha} \log \text{Tr}[\rho^{\alpha}]. \tag{4}$$

The index has been shown to be a rigid quantity in the sense that all *locally equivalent* unitary QCA, i.e. those QCA that are obtainable from each other by a finite-depth sequence of local QCA updates, have the same index. In particular, for locally generated unitary QCA, ind $= 0$, while for non-locally generated unitary QCA the index is a logarithm of a positive rational. The latter include for example the shift operation, for which the index is defined in terms of the number of shifts to the right divided by the number of shifts to the left.

## 2.3 Information current in non-unitary QCA

For non-unitary QCA, the index is no longer a rigid quantity as it does not remain invariant under local operations when coupling the system to the environment as shown by way of

$$\mathrm{Tr}\left[M_L^\dagger M_L\right] = \left(\!\!\!\!\!\!\!\!\!\!\!\right) = \left(\!\!\!\!\!\!\!\right) = \mathrm{Tr}\left[M_R^\dagger M_R\right]$$

Figure 4: Tensor network representing the equality of the traces of the first moments of $M_L^\dagger M_L$ and $M_R^\dagger M_R$. The equation holds for all QCA, independent of it being unitary or non-unitary.

several examples in Sec. 3. We seek a quantity which captures information flow in non-unitary QCA, but which is zero for local unitary QCA. This quantity should be continuous with the parameter that describes the coupling to the environment since non-unitary dynamics can be continuously connected to unitary dynamics. A natural quantity to consider, extending Eq. (1), is a continuous function on the singular values of $M_L$ and $M_R$. Note the values and the total number of non-zero singular values of $M_L$ and $M_R$ can change.

To motivate such a quantity we give a couple observations. First, the singular values of $M_L$ and $M_R$ are equal for local unitary QCA; see proof in App. A. Second, the trace of the first moments of $M_R^\dagger M_R$ and $M_L^\dagger M_L$ are equal for *all* QCA; see Fig. 4.

The lowest moment of the squared eigenvalues that can distinguish unitary and non-unitary dynamics is the second. Hence, we propose a measure of the information current, namely the information flow per update time increment, based on the Rényi-2 entropy of the operators $\sigma_\beta$ defined in Eq. (3):

$$
\begin{aligned}
I &= \frac{1}{2}(S_2(\sigma_R) - S_2(\sigma_L)) \\
&= \frac{1}{2}\log\left(\frac{\mathrm{Tr}\left[\left(M_L^\dagger M_L\right)^2\right]}{\mathrm{Tr}\left[\left(M_R^\dagger M_R\right)^2\right]}\right).
\end{aligned}
\tag{5}
$$

Thus, the current $I$ differs from the index simply by taking difference of Rényi-2 entropies rather than the Rényi-0 entropies. The tensor network description of the argument of the logarithm is shown in Fig. 5. In App. B it is shown that the current can also be reformulated in terms of the difference in Rényi-2 entropies of the inner product of the Choi-Jamiolkowski state (CJS) associated with $M_L^\dagger M_L$ and $M_R^\dagger M_R$, respectively.

In order to calculate the current we need to construct the matrices $M_L$ and $M_R$ from the QCA rule. In the following we show how to do this for local QCA using the steps graphically illustrated in Fig. 2(b-d). Here the QCA is a composition of nearest neighbour interactions between disjoint pairs of sites with a map $V$ followed by another map $W$ of the same structure but shifted by one lattice site. For the operator $V$ we can write

$$
V = \sum_{r,s=1}^{d^2} c_{r,s}\,\hat{O}_r \otimes \hat{O}_s\,,
\tag{6}
$$

where $\{\hat{O}_r\}_{r=1}^{d^2}$ is any orthonormal basis for operators on a qudit satisfying $\mathrm{Tr}\left[\hat{O}_r^\dagger \hat{O}_{r'}\right] = \delta_{r,r'}$. Note when acting on vectorized density matrices, each operator $\hat{O}_r$ acts on this doubled space as $\hat{O}_r \otimes \hat{O}_r^*$. A singular value decomposition can be performed on the matrix of coefficients

$$
c_{r,s} = \sum_{k=1}^{\chi} Y_{r,k} D_{k,k} X_{k,s}^\dagger\,,
\tag{7}
$$

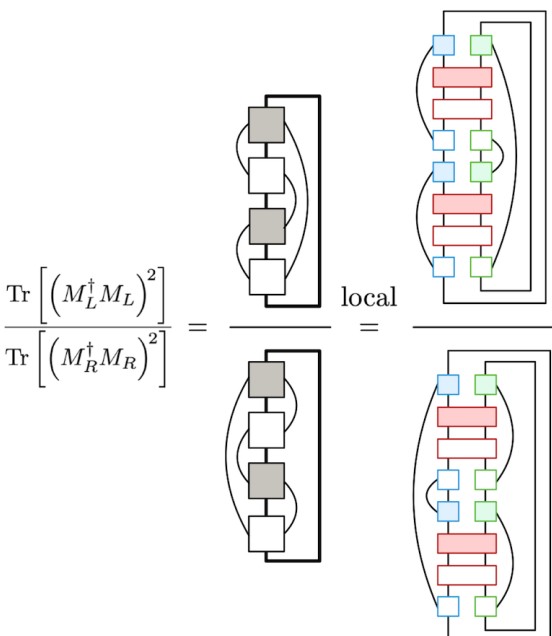

**Figure 5:** A measure of information current in QCA. The tensors are as defined in Fig. 2, where the shaded tensors represent the associated adjoints. Note that the expression on the right-hand site is only valid for local QCA.

where $Y$ and $X$ are unitary, $D$ is the diagonal matrix of singular values of $c$, and the rank of the decomposition is $\chi$ with $1 \le \chi \le d^2$.

Expanding we have

$$
\begin{aligned}
V &= \sum_{k=1}^{\chi} \sum_{r=1}^{d^2} Y_{r,k} \sqrt{D_{k,k}} \, \hat{O}_r \otimes \sum_{s=1}^{d^2} \sqrt{D_{k,k}} X^{\dagger}_{k,s} \hat{O}_s \\
&=: \sum_{k=1}^{\chi} B_k \otimes A_k \,,
\end{aligned}
\tag{8}
$$

where

$$
A_k = \sum_{s=1}^{d^2} \sqrt{D_{k,k}} \, X^{\dagger}_{k,s} \, \hat{O}_s \,, \qquad B_k = \sum_{r=1}^{d^2} \sqrt{D_{k,k}} \, Y_{r,k} \, \hat{O}_r \,.
\tag{9}
$$

Note the equally weighted symmetric distribution of the singular values on the two local matrices $A$ and $B$, with both exhibiting a factor of $\sqrt{D_{k,k}}$. This assures that the same magnitude of the current is observed after a parity operation on the QCA rule - only the sign of the current would be reversed.

Applying the singular value decomposition of $V$ according to Eqs. (6) to (9), the current can be calculated for local QCA explicitly in terms of the constituent tensors of $M_L$ and $M_R$ ($W$, $A$, and $B$, see Fig. 2(b)):

$$
I = \frac{1}{2} \log \left( \frac{\mathrm{Tr}\left[ \sum_{a,b,c,d=1}^{\chi} \left(A^{\dagger}_a \otimes B^{\dagger}_b\right) W^{\dagger} W \left(A_a \otimes B_d\right) \left(A^{\dagger}_c \otimes B^{\dagger}_d\right) W^{\dagger} W \left(A_c \otimes B_b\right) \right]}{\mathrm{Tr}\left[ \sum_{a',b',c',d'=1}^{\chi} \left(A^{\dagger}_{a'} \otimes B^{\dagger}_{b'}\right) W^{\dagger} W \left(A_{c'} \otimes B_{b'}\right) \left(A^{\dagger}_{c'} \otimes B^{\dagger}_{d'}\right) W^{\dagger} W \left(A_{a'} \otimes B_{d'}\right) \right]} \right) .
\tag{10}
$$

$$\frac{\mathrm{Tr}\left[\left(M_L^\dagger M_L\right)^2\right]}{\mathrm{Tr}\left[\left(M_R^\dagger M_R\right)^2\right]} = \frac{\phantom{xxxxx}}{\phantom{xxxxx}} = 1$$

Figure 6: On property 2.4.2: Argument of $I$ as in Fig. 5, but under the assumption of local unitary QCA such that $W^\dagger W = \mathbb{1}$. For clarity the traces are here indicated by "hooks" on the top and on the bottom of the four tensor chains, and the cyclic property of the trace has been used for the $B$'s in the numerator and the $A$'s in the denominator.

## 2.4 Properties of the information current

The main properties of the current are summarized in the points below and compared to the index for unitary QCA.

### 2.4.1 $I$ is locally computable.

As any QCA can be fully characterized by the locality-preserving operators $M$ of an MPO, and $I$ is a function of $M$, it is locally computable. The same property has been shown for the GNVW index [16].

### 2.4.2 $I$ is vanishing for unitary finite-depth circuits.

No information flow is present in the case of unitary finite-depth circuits. Mathematically, this is shown for local unitary QCA in Fig 6, where $I = \frac{1}{2}\log(1) = 0$ following Eq. (5). The pictured equation is obtained by setting $W^\dagger W = \mathbb{1} \otimes \mathbb{1} = V^\dagger V$ and using $\sum_{k=1}^{\chi} A_k A_k^\dagger = \sum_{k=1}^{\chi} B_k B_k^\dagger = \sum_{k=1}^{\chi} A_k^\dagger A_k = \sum_{k=1}^{\chi} B_k^\dagger B_k = c\mathbb{1}$; see derivation in App. A up to Eq. (A.12). In addition, note that the singular values of $M_L$ and $M_R$ are in this case equal as shown further in App. A.

The property has also been proven for the GNVW index [16].

### 2.4.3 $I$ is not invariant under blocking.

The blocking procedure describes the regrouping of all physical sites of a QCA, where two or more neighboring sites are grouped together to define a supercell. This can be viewed as a coarse-graining procedure. Here, the blocking is described by taking the tensor product of two or more local tensors of the associated MPO: $M \to M^{\otimes^n}$, where $n \in \mathbb{N}_{\geq 2}$. The corresponding tensor network description is presented in Fig. 7.

Fig. 7 shows that $I$ does not change under blocking if $W^\dagger W$ is factorizable, see condition in subfigure 7(c), while $I$ is not necessarily invariant under blocking if $W^\dagger W$ is *non*-factorizable. The latter is in distinction to the dynamics of unitary QCA which have been shown to be invariant under blocking.

However, it is possible to specify certain conditions under which the current does stay invariant under the blocking procedure. For example, if the tensor $V$ is unitary, then $I$ would remain invariant when increasing the number of blocked sites from four to larger even sized

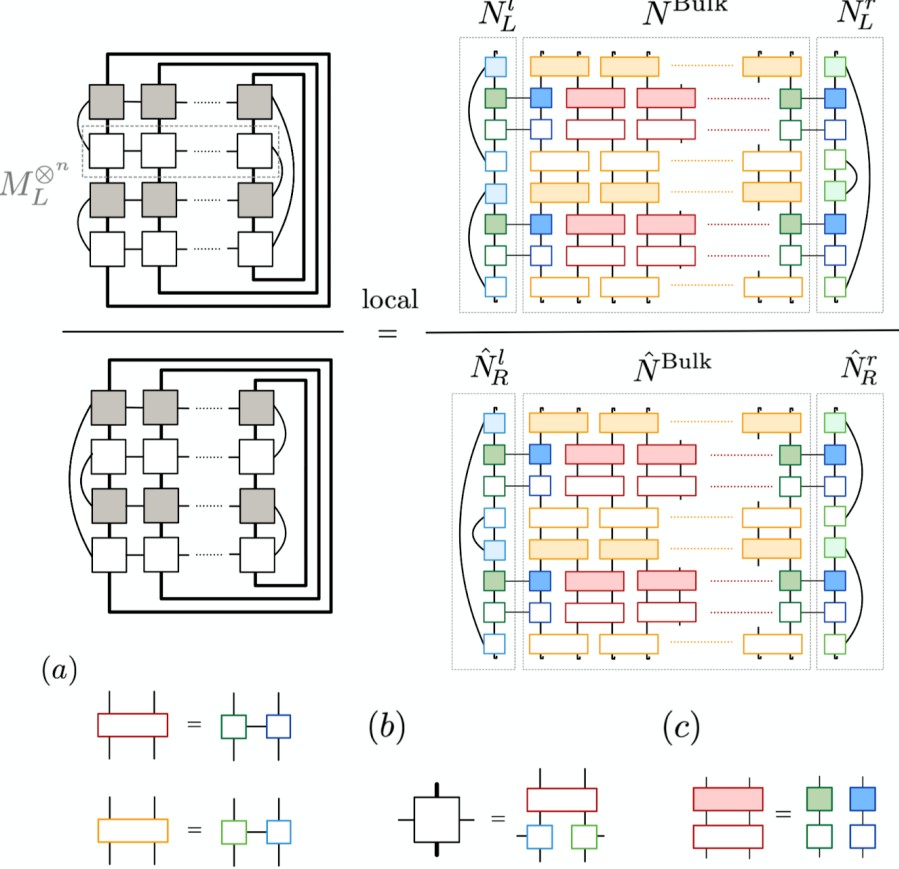

Figure 7: On property 2.4.3: Tensor network description of the blocked version of the information measure shown in Fig. 5. The expression on the left, $\dfrac{\mathrm{Tr}\left[\left(\left(M^{\otimes n}\right)_L^\dagger\left(M^{\otimes n}\right)_L\right)^2\right]}{\mathrm{Tr}\left[\left(\left(M^{\otimes n}\right)_R^\dagger\left(M^{\otimes n}\right)_R\right)^2\right]}$, holds for all QCA, whereas the one on the right, $\dfrac{\langle\hat{N}_L^l|\hat{N}^{\mathrm{Bulk}}|\hat{N}_L^r\rangle}{\langle\hat{N}_R^l|\hat{N}^{\mathrm{Bulk}}|\hat{N}_R^r\rangle}$, can only be applied to local QCA. $\hat{N}_L^l$ and $\hat{N}_R^l$ ($\hat{N}_L^r$ and $\hat{N}_R^r$) label the grouped tensors in the numerator and denominator, respectively - they act on the first (last) sites, including their associate traces and sums over their virtual indices. The composition of all other tensors acting on the physical sites in the middle are labeled by $\hat{N}^{\mathrm{Bulk}}$. Note that this tensor is the same for the numerator and the denominator, and cancel each other out if condition (c) is fulfilled: $W^\dagger W$ is factorizable (or equivalently if $\mathrm{Rank}(W^\dagger W) = 1$), such that there is no virtual bond connection between $\hat{N}^{\mathrm{Bulk}}$ and the boundary tensors. In this case the tensor diagram equals the one in Fig. 5, proving that $I$ is invariant under blocking if $W^\dagger W$ is factorizable. (a) Definition used to illustrate the singular value decomposition of the local gates $W$ (top) and $V$ (bottom). (b) Definition of the composed local tensors $M$.

regions, i.e. six, eight, ten, etc. Further, $I$ does not change by blocking for all unitary finite-depth circuits as they are factorizable with $W^\dagger W = \mathbb{1}\otimes\mathbb{1}$, see Sec. 2.4.2. There are also some unitary, not finite-depth circuits that are invariant under blocking, as for example the shift map. This is in alignment with the GNVW index introduced in Ref. [16] which has been shown to be independent of how we regroup or block sites of the unitary QCA.

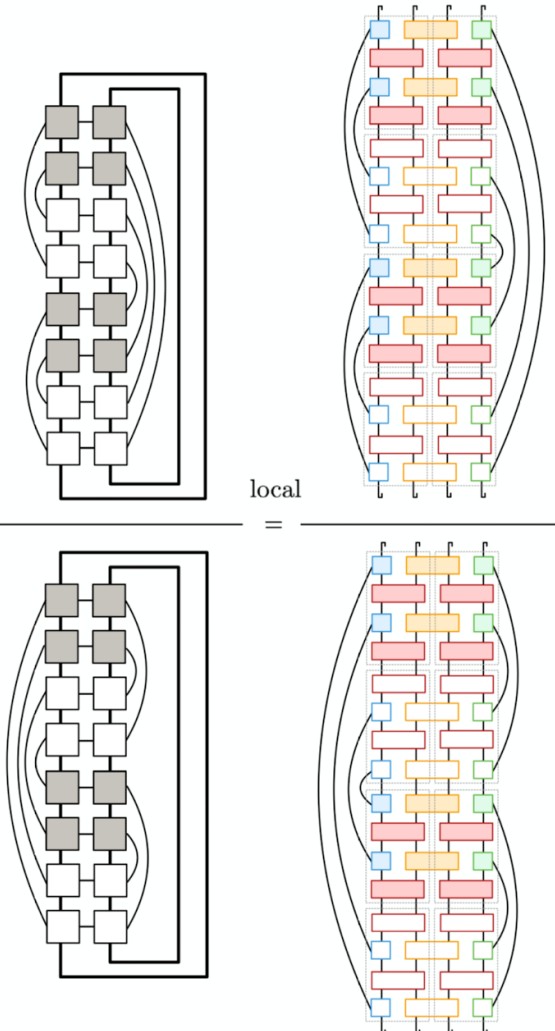

Figure 8: On property 2.4.4: Argument of the current for the composition of two QCA updates including the blocking of two sites: $\dfrac{\mathrm{Tr}\left[\left(\left(M^2\otimes M^2\right)_L^\dagger \left(M^2\otimes M^2\right)_L\right)^2\right]}{\mathrm{Tr}\left[\left(\left(M^2\otimes M^2\right)_R^\dagger \left(M^2\otimes M^2\right)_R\right)^2\right]}$. The light gray dashed boxes in the diagram on the right-hand site indicate the constituent tensors of $M^2$ (or $\left(M^2\right)^\dagger$ if the tensors are shaded).

### 2.4.4 $I$ is not additive under composition.

Similar to the blocking procedure, where physical cells are grouped together into supercells, one can also compose two or more QCA updates: $M \to \left(M^{\otimes^n}\right)^n$, where the $M^{\otimes^n}$ acts on $n$ physical sites $n$ times. To compute the current for a composition of $n$ time steps of the QCA, one has to take $n$ lattice sites into account. This can be understood by the causal cone structure of QCA that establishes a spread of information in space that is linear in time with a slope of one. Blocking insures that all sites that can contribute to the flow of information across a boundary are accounted for. Considering the composition of $n = 2$ QCA updates, the dynamics are captured in the corresponding tensor network description in Fig. 8.

The scaling of the current with the number of compositions depends on the class of QCA. For unitary finite-depth circuits, for example, the information current remains vanishing independent of the number of compositions. For the shift map, or the (classical) full-reset-SWAP QCA with $p = 1$, however, it is shown that $I$ is additive with the number of composed QCA

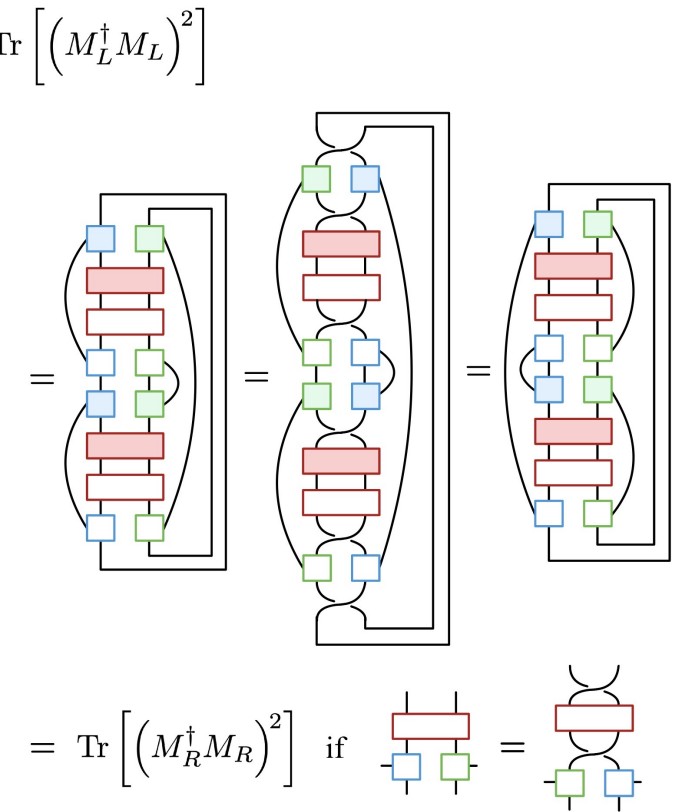

Figure 9: On property 2.4.5: Proof of $I = 0$ for swap-symmetric quantum channels, see condition on the bottom right. In the first step, the definition of $\mathrm{Tr}\left[\left(M_L^\dagger M_L\right)^2\right]$ on the top right in Fig. 5 is applied. In the second step, the swap-symmetry condition is used. In the third step, the swap operators are applied onto the (blue and green framed) tensors $A$ and $B$, which swap the physical sites they act on. In the last step, the definition of $\mathrm{Tr}\left[\left(M_R^\dagger M_R\right)^2\right]$ on the bottom right in Fig. 5 is rediscovered. The current is thus vanishing according to its definition in Eq. (5), $I = \frac{1}{2}\log(1) = 0$.

updates; see Sec. 3. This additivity under composition has also been observed for the GNVW index for all unitary QCA.[2] Other instances of QCA feature either super- and subadditivity, as discussed in Sec. 3, which show that there does not exist a general scaling of the current under composition.

### 2.4.5 *I* is vanishing if *W* and *V* are swap symmetric.

Swap-symmetric QCA are described by maps whose action is the same on both the left and the right side of the local neighborhood they act on. This means the local rule is parity symmetric and there is no preferred direction of information flow. Underpinning the intuition, the information current is shown to be vanishing for swap-symmetric QCA by the pictorial proof in Fig. 9.

---

[2]Note that in the original paper [16] it was stated that index was multiplicative (not additive) under tensoring, which has been later corrected.

Table 1: Overview of the properties of the presented exemplary maps: locality-preservation, non-unitarity, and the presence of an information current $I$. Note that example F comprises several (integrable) local and non-unitary maps which are specified in more detail in Tab. 2.

|          | A | B | C | D | E | F |
|----------|---|---|---|---|---|---|
| local    | ✗ | ✓ | ✓ | ✓ | ✓ | ✓ |
| NU       | ✗ | ✓ | ✓ | ✓ | ✓ | ✓ |
| $|I| > 0$ | ✓ | ✓ | ✓ | ✓ | ✗ | cf. Tab. 2 |

## 3 Examples

In the following, the information flow in six different types of QCA is outlined. First, the shift map in Sec. 3.1 serves as the standard example of non-local unitary QCA exhibiting non-zero information flow. Subsequently, two examples of non-unitary QCA are shown for which $W^\dagger W$ is factorizable: the reset-swap map as well as the dephase-swap map in sections 3.2 and 3.3. Next, the directed amplitude damping channel in Sec. 3.4 shows how the current behaves if $W^\dagger W$ is non-factorizable. An instance of the property 2.4.5 is provided by the asymmetric swap map in Sec. 3.5. Finally, integrable models that satisfy the Yang-Baxter equations are studied in Sec. 3.6.

All investigated maps are categorized in Tab. 1 according to whether they are local or non-local, unitary or non-unitary, and whether they exhibit a zero or non-zero current. The shift map serves as a unique example of a unitary map with non-zero current due to its non-locality property. All other listed maps are non-unitary and local.

For the quantum channels in Sec. 3.2 to Sec. 3.6, only circuits as illustrated in Fig. 2(b) are considered, where, for simplicity, the local tensors acting on the different neighborhoods at consecutive time steps are set to be the same, i.e. $W = V$. They are formally defined by superoperators which act on vectorized states $|\rho\rangle$, and exhibit an ordering of spaces $\mathcal{H}_1 \times \mathcal{H}_1^* \times \mathcal{H}_2 \times \mathcal{H}_2^*$. For this ordering of vector spaces, the map $W$ is represented as a $d^4 \times d^4$ matrix as follows:

$$W \to (\mathbb{1} \otimes \hat{\Sigma} \otimes \mathbb{1})\, W\, (\mathbb{1} \otimes \hat{\Sigma} \otimes \mathbb{1}),\tag{11}$$

where

$$\hat{\Sigma} = \sum_{j,k=0,1} |j\rangle\langle k| \otimes |k\rangle\langle j|\,,\tag{12}$$

is the swap operator which acts onto the vectorized Hilbert space.

### 3.1 Shift map (non-local unitary QCA)

The shift map is defined by a non-local unitary QCA rule that shifts the algebra uniformly to the right by one lattice site. Using the MPO description, it is defined by the local tensors of the MPO shown in Fig. 10, where the physical and virtual indices are regrouped according to left/right partitioning scheme presented in Fig. 3.

As proved diagrammatically in Fig. 11(a), the current for this map is $I = \log d^2$, which is the same as the index, $\mathrm{ind} = \log d^2$, since on the vectorized space $\mathrm{Rank}(M_R) = d^2 \cdot d^2 = d^4$ while $\mathrm{Rank}(M_L) = 1$.[3] Under coarse-graining, where two lattice sites become one, the current

---

[3]Note that the index differs from the originally defined index by a factor of two as we are working in the doubled Hilbert space.

$$M^{kj}_{kj} = \quad k \text{—}\boxempty\text{—} j$$

$$\left(M^{kj}_{kj}\right)^{\dagger} = M^{kj}_{jk} = \quad k \text{—}\blacksquare\text{—} j$$

$$\left(M^{kj}_{kj}\right)_L = M_{kk,jj} = \quad k \text{—}\boxempty\text{—} j$$

$$\left(M^{kj}_{kj}\right)^{\dagger}_L = M_{jj,kk} = \quad k \text{—}\blacksquare\text{—} j$$

$$\left(M^{kj}_{kj}\right)_R = M_{jk,kj} = \quad k \text{—}\boxempty\text{—} j$$

$$\left(M^{kj}_{kj}\right)^{\dagger}_R = M_{kj,jk} = \quad k \text{—}\blacksquare\text{—} j$$

Figure 10: The non-local shift-right QCA. Left: definitions of the relevant tensor components of $M_{L,R}$. Right: respective adjoints of the same tensor components. Superscripts define the virtual bond dimensions while subscripts represent physical bond dimensions.

does not change, see right-hand site of Fig. 11(a), whereas it is shown to be additive under an additional composition of two QCA updates, see Fig. 11(b).

## 3.2 Reset-swap map

The reset-swap map has been constructed to serve as a simple one parameter family of QCA which exhibits information flow due to its interaction with the environment which results in spatially asymmetric loss of information. Formally, it is a non-unitary quantum channel which acts on two sites: it (partially) resets either the left or the right cell, and swaps the two cells. The corresponding tensor network description for the reset-left-swap map is illustrated in Fig. 12, where, as discussed in the introduction around Fig. 1, one could already suspect that there is a directional information flow present in the system by considering the "flow" of the basis operators of the algebras which describe the input states at the individual lattice sites. One can see that there will be more operators "moving" across one side of a partition than the other.

The constituent tensors (or transfer matrices) which determine the local tensors $W$ of the associated MPO are defined in the following. The reset operation is given by the two Kraus operators

$$K_0 = \begin{pmatrix} 1 & 0 \\ 0 & \sqrt{1-p} \end{pmatrix}, \qquad K_1 = \begin{pmatrix} 0 & \sqrt{p} \\ 0 & 0 \end{pmatrix}, \tag{13}$$

which reset an input state to the $|0\rangle\langle 0|$ state with probability $p \in [0,1]$. In order to describe the corresponding action of the map onto operator algebras, the vectorization formalism in [35] is used. In this formalism, the reset gate is defined by[4]

$$R = \sum_{\mu=0,1} K_\mu \otimes K^*_\mu, \tag{14}$$

while the swap superoperator is given by

$$\text{SWAP} = \hat{\Sigma} \otimes \hat{\Sigma}^* . \tag{15}$$

[4]The vectorization follows from Eq. (70) in [35]. The conjugation of the tensors on the right-hand side of the tensor product is omitted as $\{K_\mu\}_{\mu=1,2}$ and $\hat{\Sigma}$ are real, $\hat{\Sigma} \in \mathbb{R}^{(2\times 2)}$.

(a)

$$\frac{\mathrm{Tr}\left[\left(M_L^\dagger M_L\right)^2\right]}{\mathrm{Tr}\left[\left(M_R^\dagger M_R\right)^2\right]} = \frac{}{} = \frac{}{} = \bigcirc\bigcirc = \left(d^2\right)^2$$

(b)

$$\frac{\mathrm{Tr}\left[\left(\left(M^2\otimes M^2\right)_L^\dagger \left(M^2\otimes M^2\right)_L\right)^2\right]}{\mathrm{Tr}\left[\left(\left(M^2\otimes M^2\right)_R^\dagger \left(M^2\otimes M^2\right)_R\right)^2\right]} = \frac{}{} = \bigcirc\bigcirc\bigcirc\bigcirc = \left(d^2\right)^4$$

Figure 11: (a) Derivation of the current for the shift-right QCA: $I = \frac{1}{2}\log\left(d^2\right)^2 = \log\left(d^2\right)$ remains invariant under course-graining (blocking), where two lattice sites become one. Note we work in a doubled space where each line represents the space $\mathcal{H}\times\mathcal{H}^*$, hence closed loops represent a trace which evaluates to $d^2$. Note that all the diagrams reduce to a number of closed loops. (b) Proof that under composition of $n = 2$ shifts, including the blocking of two sites, the current increases by a factor of two: $I = \frac{1}{2}\log\left(d^2\right)^4 = 2\log\left(d^2\right)$.

Then,

$$W_{1,2} = \text{SWAP}_{1,2} R_1 \,, \tag{16}$$

represents the total reset-left-swap gate, where the subscripts define the physical sites on which the associated superoperators act on. In the tensor network description used, this is the red framed tensor in Fig. 2(d). The single-site tensors $A$ and $B$ framed in blue and green are obtained by applying the SVD onto the swap superoperator as follows:

$$\text{SWAP}_{1,2} = \frac{1}{4} \sum_{a,b=0}^{3} (\sigma^a \otimes \sigma^b)_1 \otimes (\sigma^a \otimes \sigma^b)_2 \,, \tag{17}$$

where $\{\sigma^a\}_{a=0}^3 = \{I, X, Y, Z\}$ is the set of Pauli operators. Then one can write

$$\tilde{A}_1^{(a,b)} = \frac{1}{2}(\sigma^a \otimes \sigma^b)_1 \,, \qquad \tilde{B}_2^{(c,d)} = \frac{1}{2}(\sigma^c \otimes \sigma^d)_2 R_2 \,, \tag{18}$$

where the superscripts $(a, b)$ and $(c, d)$ are the virtual indices of $\tilde{A}_1$ and $\tilde{B}_2$, respectively, which arise from the SVD described above. The operators are distinguished from the above tensors $A$ and $B$ in Eq. (9) by a normalization factor that accounts for a symmetric distribution of the singular values on the two operators. This factor is henceforth ignored because it does not affect the derived result below. In total, substituting Eqs. (16) and (18) in the definition of $M$, one obtains for the local MPO tensors:

$$\begin{aligned}
M_{1,2}^{(a,b,c,d)} &= W_{1,2} \,, \left( \tilde{A}_1^{(a,b)} \otimes \tilde{B}_2^{(c,d)} \right) \\
&= \frac{1}{4} \text{SWAP}_{1,2} R_1 (\sigma^a \otimes \sigma^b)_1 \otimes (\sigma^c \otimes \sigma^d)_2 R_2 \,, \tag{19a}
\end{aligned}$$

$$\left[ M^{(a,b,c,d)} \right]^\dagger = \frac{1}{4} R_2^\dagger (\sigma^a \otimes \sigma^b)_1 \otimes (\sigma^c \otimes \sigma^d)_2 R_1^\dagger \text{SWAP}_{1,2} \,. \tag{19b}$$

Using these definitions, the derivation of the current is presented in two different ways: first, diagrammatically using the tensor network description in Fig. 13, and second, algebraically as a function of $R$ and $\{\sigma^a\}_{a=0}^3$ in App. D.

The invariance of $I$ under the blocking process is confirmed as $W^\dagger W$ is factorizable: $W^\dagger W = R_1^\dagger R_1$ due to the swap operation of the map and the fact that $R_1$ acts only non trivially (with implicit identity on the second system), see Fig. 13(a). Figure 14 shows the diagrammatic proof.

Next, Fig. 15 illustrates the non-additivity of the current under composition of two updates of the reset-swap QCA. It is shown that the argument of $I$ without composition, $\frac{\text{Tr}\left[ \left( M_L^\dagger M_L \right)^2 \right]}{\text{Tr}\left[ \left( M_R^\dagger M_R \right)^2 \right]}$, is a factor of the argument of $I$ with composition. The current of the reset-swap map thus exhibits a recursive behavior under the composition of several updates of the QCA.

A question arises regarding how the composition of the reset-swap map with a unitary finite-depth circuit would change the current. Intuitively, the information flow of the system should not change, as the current is vanishing for local unitary QCA, as proved in 2.4.2. Indeed, the current for this non-unitary QCA does not change under composition, or conjugation with a unitary, with a formal derivation presented in App. D. However, $I$ does change under composition with a unitary finite-depth circuit if two or more time steps of the QCA are composed. This is not surprising since the unitary operation can change the amount of information loss per time step.

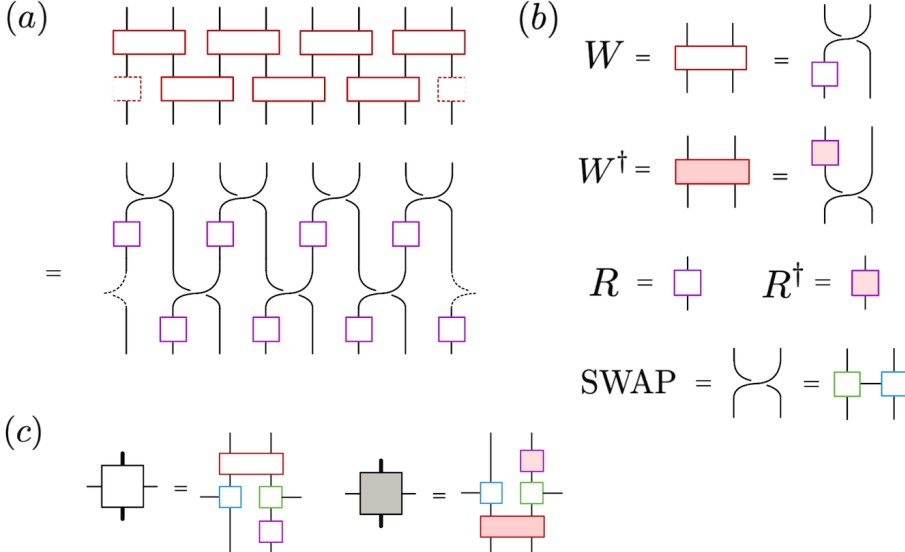

Figure 12: Illustration of the reset-swap QCA. (a) Circuit representation of one time step of the QCA. (b) Definition of the constituent local tensors $W, W^\dagger, R, R^\dagger$, and SWAP $= \sum_{k=0}^{15} A_k \otimes B_k$. (c) Definition of the total local tensors $M$ and $M^\dagger$. Note that in the representation of $M^\dagger$ the single-site tensors $A$ and $B$ are not shaded as the Pauli operators that define the swap operation are self-adjoint; $\sigma^a = (\sigma^a)^\dagger \ \forall a \in [0,3]$.

Using the definitions in Eqs. (13) to (19), a closed form expression for the current as a function of $p$ can be derived:

$$I = 2\log\left(\frac{\frac{1}{2}p^2 - p + 1}{p^2 - p + 1}\right), \tag{20a}$$

$$I_c = 2I + 2\log(\xi), \tag{20b}$$

$$\text{with} \quad \xi = \frac{(p^2 - p + 1)(p^4 - 4p^3 + 5p^2 - 2p + 1)}{p^6 - 6p^5 + 15p^4 - 19p^3 + 12p^2 - 3p + 1}.$$

The second expression $I_c$ corresponds to the current of the composition of two QCA updates. Note the current is only additive at the extremal points $p = 0$ and $p = 1$ when $\xi = 1$.

The plot of the currents $I$ and $I_c$ in Eq. (20) are shown in Fig. 16 as functions of the reset-rate $p$. One can see that the information currents for the reset-right swap and reset-left swap maps are equal and opposite. The current saturates at the same value as the index at $p = 1$. For the reset-right swap map, for $0 \le p < 1$, Rank$(M_L)$ = Rank$(M_R)$ = 16, however, at $p = 1$, Rank$(M_L)$ = 1 and Rank$(M_R)$ = 16. This gives witness to the fact that the aforementioned index experiences a jump discontinuity from ind = 0 for $0 \le p < 1$ to ind = 2 at $p = 1$. Similar behavior occurs for the reset-left swap map but where at $p = 1$, Rank$(M_L)$ = 16 and Rank$(M_R)$ = 1 and ind = $-2$. A way to interpret this is that at $p = 1$ four linearly independent operators are transported across a boundary in one direction, while only one, the $|0\rangle\langle 0|$ operator, is transported in the other direction.

## 3.3 Dephase-swap map

The dephase-swap map is similar to the reset-swap map, where the local reset gate is replaced by a dephase gate, or equivalently exchanging local amplitude damping with phase damping.

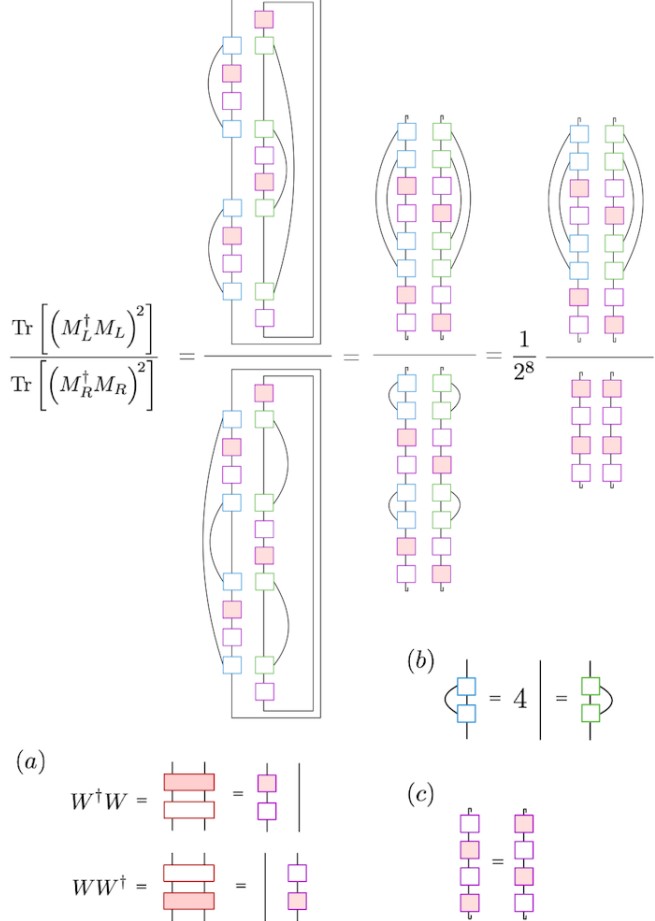

Figure 13: Computation of the current for the reset-swap QCA. The derivation follows the identities illustrated in the items below: (a) The products of two-site tensors, $W^\dagger W = R_1^\dagger \cdot \text{SWAP}_{1,2} \cdot \text{SWAP}_{1,2} \cdot R_1 = R_1^\dagger R_1$ and $W^\dagger W = R_1^\dagger \cdot \text{SWAP}_{1,2} \cdot \text{SWAP}_{1,2} \cdot R_1 = R_2 R_2^\dagger$, are separable because of the two arising consecutive swap operations. (b) The summed product of the tensors $A, B$ are trivial as these are here given by a linear combination of Pauli operators $\frac{1}{4} \sum_{a,b=0}^{3} (\sigma^a \otimes \sigma^b)(\sigma^a \otimes \sigma^b) = 4\mathbb{1}$. (c) The cyclic property of the trace is used, $\text{Tr}\left[R^\dagger R R^\dagger R\right] = \text{Tr}\left[R R^\dagger R R^\dagger\right]$.

The Kraus operators describing the dephase operation are:

$$K_0 = \begin{pmatrix} \sqrt{1-p/2} & 0 \\ 0 & \sqrt{1-p/2} \end{pmatrix},$$

$$K_1 = \begin{pmatrix} \sqrt{p/2} & 0 \\ 0 & -\sqrt{p/2} \end{pmatrix}, \tag{21}$$

with $p \in [0,1]$. For this map, when acting on the left site for $0 \le p < 1$, $\text{Rank}(M_L) = \text{Rank}(M_R) = 16$, while for $p = 1$, $\text{Rank}(M_L) = 4$ and $\text{Rank}(M_R) = 16$. Hence, similar to the case for this reset-swap map, the index experiences a jump discontinuity from $\text{ind} = 0$ for $0 \le p < 1$ to $\text{ind} = 1$ at $p = 1$. Similar behavior occurs for the reset-left swap map but where at $p = 1$, $\text{Rank}(M_L) = 16$ and $\text{Rank}(M_R) = 4$ and $\text{ind} = -1$. Like the reset-swap map, at $p = 1$, in one direction four linearly independent operators are transported across a boundary, but in contrast, two operators, $|0\rangle\langle 0|$ and $|1\rangle\langle 1|$, are transported in the other direction.

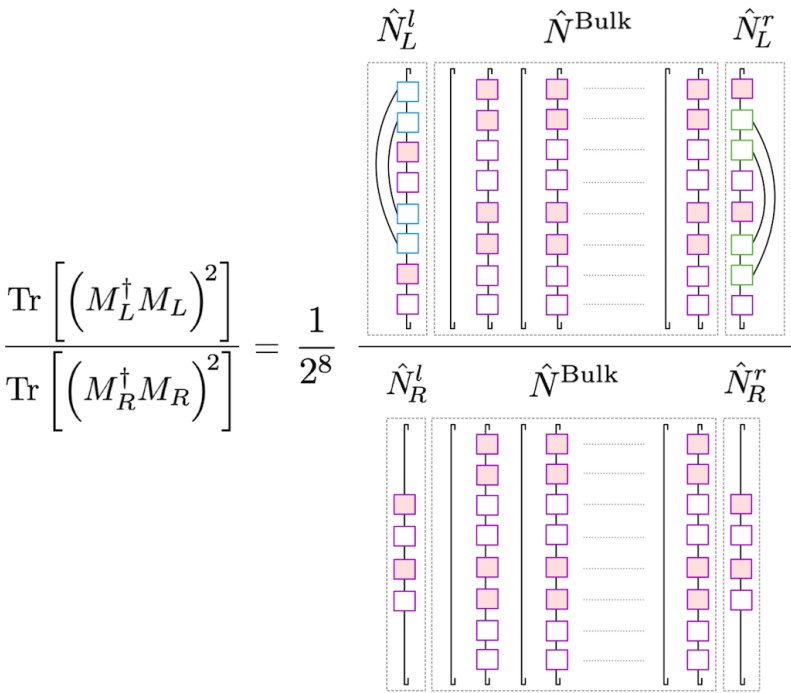

$$\frac{\mathrm{Tr}\left[\left(M_L^\dagger M_L\right)^2\right]}{\mathrm{Tr}\left[\left(M_R^\dagger M_R\right)^2\right]} = \frac{1}{2^8}$$

Figure 14: Invariance of the current under blocking for the coarse-grained reset-swap QCA, shown by the tensor network description of the argument of $I$. Note the marked terms $\hat{N}^{\mathrm{Bulk}}$ are the same for the numerator and the denominator and cancel out. The terms on the very left and right sites of the supercell, $\{\hat{N}_L^l, \hat{N}_L^r, \hat{N}_R^l, \hat{N}_R^r\}$ including the prefactor, equal the expression for the argument of $I$ associated with the non-coarse-grained QCA, see right-hand site in Fig. 13.

The solution of $I$ for the dephase-left swap map without and with composition (and blocking) of two QCA updates:

$$I = \log\left(\frac{2\left(\frac{1}{2}p^2 - p + 1\right)^2}{(p^4 - 4p^3 + 6p^2 - 4p + 2)}\right), \tag{22a}$$

$$I_c = 2I + 2\log(\xi), \tag{22b}$$

$$\text{with} \quad \xi = \frac{(p^4 - 4p^3 + 5p^2 - 2p + 1)^2}{p^8 - 8p^7 + 28p^6 - 56p^5 + 69p^4 - 52p^3 + 22p^2 - 4p + 1},$$

where the corresponding plots are presented in Fig. 16. Again, the current is only additive at the points $p = 0, 1$. The information currents for the dephase-right swap and dephase-left swap maps are equal and opposite, whereby the current saturates at the same value as the index at $p = 1$.

## 3.4 Directed amplitude damping map

In this section, the information flow of a QCA is investigated whose local tensors do not separate like $W^\dagger W = \tilde{W}_1 \otimes \tilde{W}_2$, where $\tilde{W}_1, \tilde{W}_2$ are arbitrary tensors acting on the associated left or the right site of a two-cell subsystem.

$$\frac{\phantom{XXXXXX}}{\phantom{XXXXXX}} = \frac{\phantom{XXXXXX}}{\phantom{XXXXXX}} = \frac{1}{2^8} \frac{\phantom{XXXXXX}}{\phantom{XXXXXX}} \times \frac{\mathrm{Tr}\left[\left(M_L^\dagger M_L\right)^2\right]}{\mathrm{Tr}\left[\left(M_R^\dagger M_R\right)^2\right]}$$

Figure 15: Argument of the current for two updates of the reset-swap QCA, including the blocking of two physical sites. The derivation is analogous to the derivation of the current without composition in Fig. 13, but shows the swap operations in the first step (which are in Fig. 13 omitted for clarity).

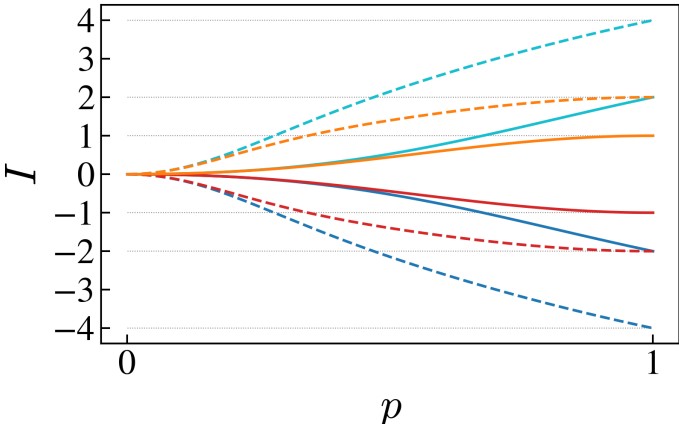

Figure 16: Information current $I$ for the reset-left-swap (blue), reset-right-swap (cyan), dephase-left-swap (red), and dephase-right-swap (orange) map as a function of the reset- or dephase rate $p$, where dashed lines indicate the composition of two time steps of the associated QCA.

The quantum channel is described by a directed amplitude damping map from the $|01\rangle$ to the $|00\rangle$ state with probability $p \in [0,1]$:

$$
\begin{aligned}
|00\rangle_S |0\rangle_E &\to |00\rangle_S |0\rangle_E \,, \\
|01\rangle_S |0\rangle_E &\to \sqrt{1-p}\,|01\rangle_S |0\rangle_E + \sqrt{p}\,|00\rangle_S |1\rangle_E \,, \\
|10\rangle_S |0\rangle_E &\to |10\rangle_S |0\rangle_E \,, \\
|11\rangle_S |0\rangle_E &\to |11\rangle_S |0\rangle_E \,,
\end{aligned}
$$

where the subscripts $S$ and $E$ indicate the system or the environmental subsystem, respectively. Tracing out the environment $E$ results in a quantum channel acting on (solely) the system $S$ defined by the Kraus operators

$$
K_0 = \begin{pmatrix} 1 & 0 & 0 & 0 \\ 0 & \sqrt{1-p} & 0 & 0 \\ 0 & 0 & 1 & 0 \\ 0 & 0 & 0 & 1 \end{pmatrix}, \qquad K_1 = \begin{pmatrix} 0 & \sqrt{p} & 0 & 0 \\ 0 & 0 & 0 & 0 \\ 0 & 0 & 0 & 0 \\ 0 & 0 & 0 & 0 \end{pmatrix}, \tag{23}
$$

that describe discrete-time dynamics and define the CPTP map in the considered vectorization formalism.[5]

The channel could alternatively be generated by the continuous-time Lindblad dynamics described by the Master equation

$$
\rho(t+\tau) = e^{\mathcal{L}\tau}[\rho(t)], \tag{25}
$$

which is defined by the Liouvillian

$$
\mathcal{L}[\rho] = \left( L\rho L^\dagger - \frac{1}{2}\left(L^\dagger L\rho + \rho L^\dagger L\right) \right), \tag{26}
$$

---

[5]The definition of the map in its vectorized form follows from Eq. (70) in [35]. The conjugation of the tensors on the right-hand side of the tensor product can be omitted because $\{K_\mu\}_{\mu=1,2}$ are in this case real.

$$
V = \sum_{\mu=0,1} K_\mu \otimes K_\mu^*. \tag{24}
$$

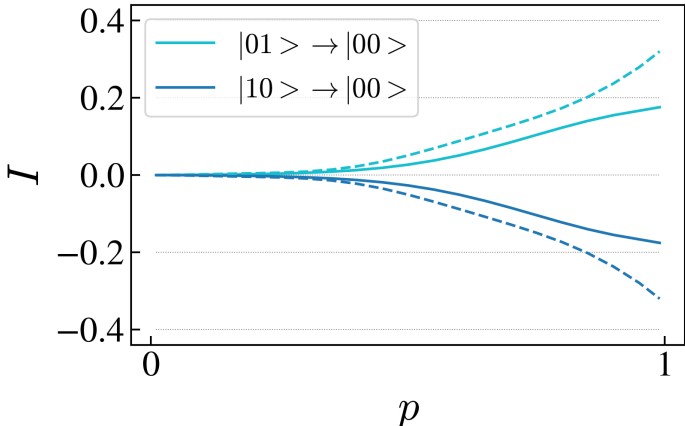

Figure 17: Information current $I$ for a QCA that describes directed amplitude damping as labeled, where $p$ is the amplitude damping rate. Dashed lines correspond to the composition two time steps of the QCA.

with jump operator

$$L = \sqrt{-\frac{1}{\tau}\ln(1-p)}\,|00\rangle\langle 01|\,, \tag{27}$$

where the time of the action of the channel is set to $\tau = 1$ for simulating one corresponding discrete time step.

Similarly, the map that describes directed amplitude damping in the opposite direction, from the $|10\rangle$ to the $|00\rangle$ state, is:

$$|00\rangle_S\,|0\rangle_E \to |00\rangle_S\,|0\rangle_E\,,$$
$$|01\rangle_S\,|0\rangle_E \to |01\rangle_S\,|0\rangle_E\,,$$
$$|10\rangle_S\,|0\rangle_E \to \sqrt{1-p}\,|10\rangle_S\,|0\rangle_E + \sqrt{p}\,|00\rangle_S\,|1\rangle_E\,,$$
$$|11\rangle_S\,|0\rangle_E \to |11\rangle_S\,|0\rangle_E\,,$$

where Kraus operators are given by

$$K_0 = \begin{pmatrix} 1 & 0 & 0 & 0 \\ 0 & 1 & 0 & 0 \\ 0 & 0 & \sqrt{1-p} & 0 \\ 0 & 0 & 0 & 1 \end{pmatrix}, \qquad K_1 = \begin{pmatrix} 0 & 0 & \sqrt{p} & 0 \\ 0 & 0 & 0 & 0 \\ 0 & 0 & 0 & 0 \\ 0 & 0 & 0 & 0 \end{pmatrix}, \tag{28}$$

with corresponding jump operator

$$L = \sqrt{-\frac{1}{\tau}\ln(1-p)}\,|00\rangle\langle 10|\,. \tag{29}$$

The resulting current is in Fig. 17 plotted as a function of the damping rate $p \in [0, 1]$. As expected, the current increases monotonically with the damping rate, and no additivity under composition is observed. Additionally, this is a first example we have described so far where $W^\dagger W$ is not separable, see proof in App. E. The current is therefore not invariant under blocking.

### 3.5   Asymmetric swap map

An asymmetric swap operation acting on two physical sites is investigated next. The asymmetry is realized by considering a partial swap gate: the $|01\rangle$ state is mapped to the swapped $|10\rangle$ state with a certain probability $p_{01} \in [0,1]$, where the latter is not necessarily equal to the transition probability $p_{10} \in [0,1]$ for the reverse process from the $|10\rangle$ to the $|01\rangle$ state:

$$
\begin{aligned}
|00\rangle_S |0\rangle_E &\to |00\rangle_S |0\rangle_E \,, \\
|01\rangle_S |0\rangle_E &\to \sqrt{1-p_{01}}\,|01\rangle_S |0\rangle_E + \sqrt{p_{01}}\,|10\rangle_S |1\rangle_E \,, \\
|10\rangle_S |0\rangle_E &\to \sqrt{1-p_{10}}\,|10\rangle_S |0\rangle_E + \sqrt{p_{10}}\,|01\rangle_S |1\rangle_E \,, \\
|11\rangle_S |0\rangle_E &\to |11\rangle_S |0\rangle_E \,.
\end{aligned}
$$

By tracing out the environmental subsystem, a quantum channel acting on the system is defined by the Kraus operators

$$
K_0 = \begin{pmatrix}
1 & 0 & 0 & 0 \\
0 & \sqrt{1-p_{01}} & 0 & 0 \\
0 & 0 & \sqrt{1-p_{10}} & 0 \\
0 & 0 & 0 & 1
\end{pmatrix},
$$

$$
K_1 = \begin{pmatrix}
0 & 0 & 0 & 0 \\
0 & 0 & \sqrt{p_{10}} & 0 \\
0 & \sqrt{p_{01}} & 0 & 0 \\
0 & 0 & 0 & 0
\end{pmatrix}. \tag{30}
$$

The continuous-time version of this map is generated by the Liouvillian

$$
\mathcal{L}[\rho] = \sum_{k=1,2} \left( L^{(k)} \rho L^{(k)^\dagger} - \frac{1}{2} \left( L^{(k)^\dagger} L^{(k)} \rho + \rho L^{(k)^\dagger} L^{(k)} \right) \right), \tag{31}
$$

which includes two jump operators

$$
\begin{aligned}
L^{(1)} &= \sqrt{-\frac{1}{\tau} \ln(1-p_{01})}\,|10\rangle\langle 01| \,, \\
L^{(2)} &= \sqrt{-\frac{1}{\tau} \ln(1-p_{10})}\,|01\rangle\langle 10| \,,
\end{aligned} \tag{32}
$$

that are each associated with the damping of the $|01\rangle$ or the $|10\rangle$ state, respectively.

In total, no information flow is observed, $I = 0$, despite the asymmetry of the mapping when $p_{01} \neq p_{10}$. This meets expectations nonetheless because of the swap-symmetry of the map. Namely, we observe: (1) conjugation of the map with a unitary can be applied w.l.o.g. as it does not change the current, and (2) choosing to conjugate with the unitary $X_1 X_2$, the jump operators satisfy $\forall\, p_{01} \neq p_{10}$:

$$
\begin{aligned}
\mathrm{SWAP}_{1,2}\,(X_1 X_2)\,L^{(1)}\,(X_1 X_2)\,\mathrm{SWAP}_{1,2} &= L^{(1)} \,, \\
\mathrm{SWAP}_{1,2}\,(X_1 X_2)\,L^{(2)}\,(X_1 X_2)\,\mathrm{SWAP}_{1,2} &= L^{(2)} \,,
\end{aligned} \tag{33}
$$

which establishes the swap symmetry. The proof of the invariance of the current under the swap operation is given in Fig. 9.

Table 2: Overview of the properties of the integrable models A1, A2, B1, B2, and B3 presented in [37]. The CPTP maps have been formulated and categorized in [37] according to whether they are unital, lower triangular and/or conserve $S^z$. Three out of the five models are in this work shown to have a non-zero information current $I$ and notably this feature is independent of the unital property, the conservation of $S^z$, and whether the superoperator is lower triangular.

|                  | A1 | A2 | B1 | B2 | B3 |
|------------------|----|----|----|----|----|
| unital           | ✗  | ✗  | ✓  | ✓  | ✗  |
| $\Delta S^z = 0$ | ✗  | ✗  | ✓  | ✓  | ✓  |
| lower triangular | ✓  | ✓  | ✗  | ✗  | ✗  |
| $|I| > 0$        | ✓  | ✓  | ✗  | ✓  | ✗  |

### 3.6 Integrable non-unitary maps

Recently there has been an investigation of one-dimensional integrable non-unitary QCA [36, 37]. These are models where the open systems dynamics are of Lindblad type and the maps generated by each Lindblad summand define a non-unitary map that is a solution to the Yang-Baxter equation.

A classification of integrable Lindbladians are given in [36] and their corresponding CPTP maps are presented in [37]. All their listed models (A1, A2, B1, B2, B3) are specified below in terms of their CPTP map representation, and the corresponding information currents $I$ are inspected.

The models studied in 2 come in two classes. The models of A type have Lindblad jump operators which are lower triangular with at most two elements below the diagonal in the logical basis, and they are non-unital maps. Model A1 has one element below the diagonal, and model A2 has two elements below the diagonal. The models of B type conserve total spin projection along the $\hat{z}$ axis, $S^z = \frac{1}{2} \sum_j Z_j$, and only models B1 and B2 are unital. A summary of the properties of these models is give in Table 2.

#### 3.6.1 Models A1 and A2

The CPTP maps of models A1 and A2 in [37] are investigated at first. Model A1 is in the Kraus representation given by the two Kraus operators

$$K_0 = e^{-u/2} \begin{pmatrix} 1 & 0 & 0 & 0 \\ 0 & e^{-u/2} & 0 & 0 \\ 0 & 0 & e^{u/2} & 0 \\ 0 & 0 & 0 & 1 \end{pmatrix}, \tag{34a}$$

$$K_1 = \sqrt{1 - e^{-u}} \begin{pmatrix} 1 & 0 & 0 & 0 \\ 0 & e^{-u/2} & 0 & 0 \\ 0 & -ie^{i\phi} & 0 & 0 \\ 0 & 0 & 0 & 1 \end{pmatrix}, \tag{34b}$$

where $0 \le u \in \mathbb{R}$ and $\phi \in \mathbb{R}$ represents an arbitrary phase factor. Due to the phase factor in $K_1$ this is the first map we encounter that includes complex numbers.

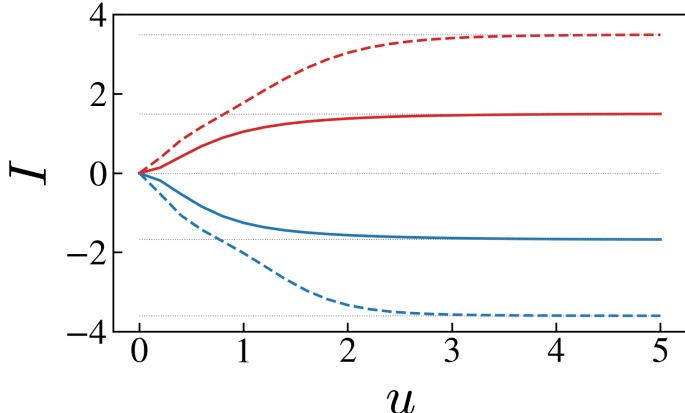

Figure 18: Information current $I$ corresponding to the integrable channels given in models A1 (blue) and A2 (red) in [37], see Kraus operators in Eqs. (34) and (35), respectively. Note that in calculations of map A1, the choice of the phase factor $\phi \in \mathbb{R}$ does not change $I$, while for model A2, the current does analogously turn out to be independent of the choice of the sign of $\tau = \pm 1$. The current $I_c$ associated with the composition of two time steps of the corresponding QCA is represented by dashed lines in the same color as the model.

Model A2 is defined by the Kraus operators

$$K_0 = e^{-u} \begin{pmatrix} 1 & 0 & 0 & 0 \\ 0 & e^{-u/2} & 0 & 0 \\ 0 & -\tau(e^u - 1) & e^{u/2} & 0 \\ 0 & 0 & 0 & e^u \end{pmatrix}, \tag{35a}$$

$$K_1 = \sqrt{e^u - 1}\, e^{-u} \begin{pmatrix} 1 & 0 & 0 & 0 \\ 0 & 0 & 0 & 0 \\ 0 & \tau & e^{u/2} & 0 \\ e^{u/2} & 0 & 0 & 0 \end{pmatrix}, \tag{35b}$$

where $\tau = \pm 1$.

The associated currents $I$ are plotted as a function of $u \geq 0$ in Fig. 18. One can see that for both maps $|I|$ is monotonically increasing with $u$ and saturates at about $u = 3$ to a constant value: $I = -1.68$ and $I_c = -3.61$ for model A1, and $I = 1.50$ and $I_c = 3.49$ for model A2.

### 3.6.2 Model B1

The CPTP map of model B1 in [37] is expressed by

$$K_0 = \sqrt{\frac{1}{u+1}}\, \mathbb{1}_{4\times 4}, \qquad K_1 = \sqrt{\frac{u}{u+1}} \begin{pmatrix} 1 & 0 & 0 & 0 \\ 0 & 0 & \tau & 0 \\ 0 & \tau & 0 & 0 \\ 0 & 0 & 0 & \kappa\tau \end{pmatrix}, \tag{36}$$

where $u \geq 0$, $\tau, \kappa = \pm 1$, and $\mathbb{1}_{4\times 4}$ represents the identity channel acting on two qubits.

Because of the swap symmetry of this channel (36), the current is vanishing, $I = 0$; see proof in Sec. 2.4.5.

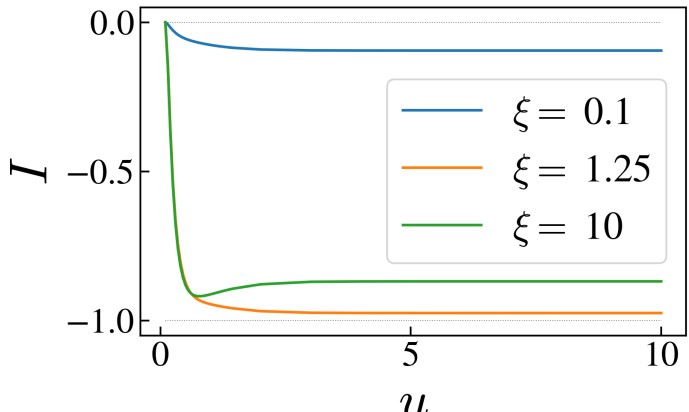

Figure 19: Information current $I$ for the integrable CPTP map given in model B2 in [37]. The map is given by the Kraus operators in Eq. (37), where for the presented calculations the parameters $v = 0.1$ and $\phi = 0$ are chosen. Due to the above condition $u \geq v$, the plot starts at $u = v = 0.1$.

### 3.6.3 Model B2

Model B2 is defined by the Kraus operators

$$K_0 = \sqrt{\beta}\, G(u-v), \; K_1 = \sqrt{\alpha}\, G(u+v)(Z \otimes \mathbb{1}_{2\times 2}),\tag{37}$$

where

$$G(u) = \frac{1}{\cosh(u)}\begin{pmatrix} \cosh(u) & 0 & 0 & 0 \\ 0 & 1 & -i\sinh(u)e^{i\phi} & 0 \\ 0 & -i\sinh(u)e^{-i\phi} & 1 & 0 \\ 0 & 0 & 0 & \cosh(u) \end{pmatrix},\tag{38}$$

$\phi \in [0, 2\pi)$ is an arbitrary phase, and $\mathbb{1}_{2\times 2}$ labels the identity channel acting on one qubit. The normalization constants $\sqrt{\beta}$ and $\sqrt{\alpha}$ of each, $K_0$ or $K_1$, respectively, are given by

$$\beta = \frac{\cosh(u-v)\cosh(\xi - \eta)}{\cosh(u-v)\cosh(\xi - \eta) + \cosh(u+v)\sinh(\xi - \eta)},\tag{39}$$

and $\alpha = 1 - \beta$ under the conditions that $u \geq v$ and

$$\frac{\sinh(2\xi)}{\sinh(2u)} = \frac{\sinh(2\eta)}{\sinh(2v)}.\tag{40}$$

The associated current is shown below in Fig. 19.

Note that $I = 0$ if $u = v = 0.1$ due to the unitarity of the map is in this case. For all $u \geq v$, $I$ is decreasing (monotonically for small $\xi$) and saturates to a constant value for all about $u \geq 3$: $I = -0.095$ for $\xi = 0.1$, $I = -0.97$ for $\xi = 1.25$, and $I = -0.87$ for $\xi = 10$. The constant $\xi = 1.25$ has been chosen because the absolute value of the current is for this value maximum, given $v = 0.1$.

For $u \to \infty$ the current is observed to be vanishing due to the unitarity of the map in this limit. If one were to set $\xi = 0$, then one would find with Eq. (40) that $\xi = \eta$, leading to $\beta = 1$ with Eq. (39) and $\alpha = 1 - \beta = 0$, such that $K_0 = G(u-v)$ and $K_1 = 0$ according to Eq. (37) — the channel is therefore unitary and $I$ vanishes for $\xi = 0$.

It is notable that the computational complexity of calculating the current for this model is drastically increased in comparison to the previously presented models. Here, the ranks of the considered local tensors are equal to 16 for one time step, or $16^2 = 256$ for two

times steps, in comparison to 4 or 16, respectively, in models A1 and A2. Multiplying the rank, which is the dimension of the virtual degree of freedom of the MPO tensors, with the dimension of the (vectorized) physical degree of freedom, $4^2 = 16$ for two lattice sites or $4^4 = 256$ for four lattice sites, one obtains the size of the matrices $M_L$ and $M_R$: $(256 \times 256)$ for a single time step, or $(65\,536 \times 65\,536)$, i.e. squared, for two composed updates of the associated QCA. Because the calculation of the information current $I$ involves more complex matrix transformations that are computationally more costly for matrices of the latter size, additional plots for the current $I_c$ of two QCA time steps are not performed here. However, the value $I_c(u = 5, v = 0.1, \xi = 1.25, \phi = 0) = -1.949669$ has been calculated, which, interestingly, is very close (up to an error of $10^{-5}$) to twice the current without the composition: $2I(u = 5, v = 0.1, \xi = 1.25, \phi = 0) = -1.949698$.[6] Whether this near additivity occurs for other parameter regimes would require further study.

### 3.6.4 Model B3

Model B3 is given by the Kraus operators

$$K_0 = \frac{1}{\eta}\begin{pmatrix} \eta & 0 & 0 & 0 \\ 0 & 1 & -i(1-\gamma)\sinh(\alpha)e^{i\phi} & 0 \\ 0 & -i(1+\gamma)\sinh(\alpha)e^{-i\phi} & 1 & 0 \\ 0 & 0 & 0 & \eta \end{pmatrix}, \tag{41a}$$

$$K_1 = \frac{\xi}{\eta}\begin{pmatrix} 0 & 0 & 0 & 0 \\ 0 & -e^{\alpha}(1-\gamma) & i(1-\gamma)e^{i\phi} & 0 \\ 0 & i(1+\gamma)e^{-i\phi} & e^{\alpha}(1+\gamma) & 0 \\ 0 & 0 & 0 & 0 \end{pmatrix}, \tag{41b}$$

where $\gamma \geq 0$, $\alpha = (\gamma^2+1)u/2$, $\eta = \gamma\sinh(\alpha)+\cosh(\alpha)$, and $\xi = \sqrt{\gamma\sinh(\alpha)/[(1+\gamma^2)\cosh(\alpha)+2\gamma\sinh(\alpha)]}$.

The corresponding current has been found to be vanishing for all $u \geq 0$ and $\gamma \geq 0$.

## 4 Conclusion

We have introduced a measure of information flow for open quantum system dynamics which captures features that are not obtained from the unitary case. Considering the MPO representation of QCA, the information current has been constructed using the left or right partitionings of the local tensors of the MPO, similar to the description of the index theory using MPUs [28, 29]. While the prior studied index can be expressed in terms of the difference of Rényi-0 entropies of the singular values associated with the left and right partitionings of these tensors, the current, is defined by the difference in Rényi-2 entropies of the corresponding singular values.

The current is locally computable, vanishing for finite-depth unitary circuits and SWAP-symmetric QCA, and is continuous in the noise parameters defining the maps. It has been shown to exhibit the same value as the index for the shift map, the standard example of a nonlocal unitary QCA. In contrast to the index, the information current is not generically invariant under blocking of lattice sites, nor is it additive under composition of QCA updates. Failure to be additive under composition is not surprising given the nonlinearity of the function we are evaluating. The general lack of invariance under blocking indicates that bulk properties of the (blocked) supercells can change the information flow at the edges of the cell, though there is

---

[6]The scaling factor could be a result of $I$ not being fully saturated for $u = 5$, see plot in Fig. 19, but where we leave this question for further research.

a large class of open systems dynamics we have identified (when $W^\dagger W$ is factorizable) where the information current is invariant under blocking.

A particularly interesting use case for our measure are the integrable models in Sec. 3.6, which have been shown to exhibit a particle flow in certain cases; see Eq. (20) in [36]. As the information current $I$ is independent of the type of particle and the choice of basis, our measure provides a generalization to the notion of a particle flow in [36].

One might ask whether the information current for non-unitary QCA has a thermodynamic interpretation wherein a spatially periodic heat exchange between the environment and the system generates a net information current. For example, in the case of the reset-swap map, by Landauer's principle, the reset map necessarily involves a heat exchange of $k_B T \ln 2$ between every second site and the environment, where $k_B$ is the Boltzmann constant and $T$ is the temperature of the environment. We leave this matter to future work.

Lastly, it is noted that the implementation of non-unitary QCA can be realized using lattices of ultracold atoms excited to Rydberg states [38]. Radius one rules in this implementation can be used for a variety of tasks including dissipative entangled resource state preparation and the study of non-equilibrium phase transitions [39–41]. The information current could be indirectly measured in such a setup by performing process tomography using a quorum of properly prepared input states and measurements on output states of spins in neighborhood of the QCA rule. This would enable determination of the positive Hermitian operators $\sigma_\beta = M_\beta^\dagger M_\beta / \text{Tr}\left[M_\beta^\dagger M_\beta\right] \; \forall \beta \in \{L, R\}$ in Eq. (3) from which the current could be calculated.

# Acknowledgments

We acknowledge helpful discussions with David Gross.

**Funding information** This research was supported by the Australian Research Council Centre of Excellence for Engineered Quantum Systems (EQUS, CE170100009), by the Sydney Quantum Academy, Sydney, NSW, Australia, and with the assistance of resources from the National Computational Infrastructure (NCI), which is supported by the Australian Government.

# A  Singular values of $M_L$ and $M_R$ of local unitary QCA

It is proved that the singular values of $M_L$ an $M_R$ are equal for local unitary QCA. The contractions giving the traces of the second moments of $M_L^\dagger M_L$ and $M_R^\dagger M_R$ are shown in Fig. 6. If the QCA rule is unitary, then $W^\dagger W = \mathbb{1}$, and as is apparent from the diagram,

$$\text{Tr}\left[\left(M_L^\dagger M_L\right)^2\right] = \text{Tr}\left[\left(\sum_{k=1}^{\chi} A_k^\dagger A_k\right)^2\right] \text{Tr}\left[\left(\sum_{k=1}^{\chi} B_k B_k^\dagger\right)^2\right], \tag{A.1a}$$

$$\text{Tr}\left[\left(M_R^\dagger M_R\right)^2\right] = \text{Tr}\left[\left(\sum_{k=1}^{\chi} A_k A_k^\dagger\right)^2\right] \text{Tr}\left[\left(\sum_{k=1}^{\chi} B_k^\dagger B_k\right)^2\right]. \tag{A.1b}$$

For unitary QCA, the tensor $V$ is also unitary. From Eq. (9), the local tensors are

$$B_k = \sum_{r=1}^{d^2} \sqrt{D_{k,k}} Y_{r,k} \hat{O}_r, \qquad A_k = \sum_{s=1}^{d^2} \sqrt{D_{k,k}} X_{k,s}^\dagger \hat{O}_s. \tag{A.2}$$

Using the fact that $V^\dagger V = \mathbb{1} \otimes \mathbb{1}$ we have

$$
\begin{aligned}
\mathbb{1} \otimes \mathbb{1} &= \sum_{k,k'=1}^{\chi} B_k^\dagger B_{k'} \otimes A_k^\dagger A_{k'} \\
&= \sum_{k,k'=1}^{\chi} D_{k,k} D_{k',k'} \left( \sum_{r,r'=1}^{d^2} Y_{r,k}^* Y_{r',k'} O_r^\dagger O_{r'} \otimes \sum_{s,s'=1}^{d^2} (X^\dagger)_{k,s}^* X_{k',s'}^\dagger O_s^\dagger O_{s'} \right).
\end{aligned} \tag{A.3}
$$

Taking a partial trace over the second factor, i.e. on Hilbert space $\mathcal{H}_2 \times \mathcal{H}_2^*$, gives

$$
\begin{aligned}
d^2 \mathbb{1} &= \sum_{k,k'=1}^{\chi} D_{k,k} D_{k',k'} \sum_{r,r'=1}^{d^2} Y_{r,k}^* Y_{r',k'} O_r^\dagger O_{r'} \sum_{s,s'=1}^{d^2} (X^\dagger)_{k,s}^* X_{k',s'}^\dagger \operatorname{Tr}\!\left[ O_s^\dagger O_{s'} \right] \\
&= \sum_{k,k'=1}^{\chi} D_{k,k} D_{k',k'} \sum_{r,r'=1}^{d^2} Y_{r,k}^* Y_{r',k'} O_r^\dagger O_{r'} \sum_{s,s'=1}^{d^2} X_{k',s'}^\dagger X_{s,k} \delta_{s,s'} \\
&= \sum_{k=1}^{\chi} D_{k,k}^2 \sum_{r,r'=1}^{d^2} Y_{r,k}^* Y_{r',k} O_r^\dagger O_{r'} \\
&= \sum_{k=1}^{\chi} D_{k,k} B_k^\dagger B_k.
\end{aligned} \tag{A.4}
$$

Taking a second trace gives

$$
\sum_{k=1}^{\chi} D_{k,k}^2 = d^4. \tag{A.5}
$$

Similarly, taking the partial trace on the first factor, i.e. on Hilbert space $\mathcal{H}_1 \times \mathcal{H}_1^*$, gives

$$
d^2 \mathbb{1} = \sum_{k=1}^{\chi} D_{k,k} A_k^\dagger A_k. \tag{A.6}
$$

An analogous argument using $VV^\dagger = \mathbb{1}$ shows that

$$
\sum_{k=1}^{\chi} D_{k,k} B_k B_k^\dagger = \sum_{k=1}^{\chi} D_{k,k} A_k A_k^\dagger = d^2 \mathbb{1}. \tag{A.7}
$$

Additionally,

$$
\begin{aligned}
\operatorname{Tr}\!\left[ \sum_{k=1}^{\chi} A_k^\dagger A_k \right] &= \sum_{k=1}^{\chi} \sum_{s,s'=1}^{d^2} D_{k,k} (X^\dagger)_{k,s}^* X_{k,s'}^\dagger \operatorname{Tr}\!\left[ \hat{O}_s^\dagger \hat{O}_{s'} \right] \\
&= \sum_{k=1}^{\chi} D_{k,k} \sum_{s=1}^{d^2} X_{k,s}^\dagger X_{s,k} \\
&= \operatorname{Tr}[D],
\end{aligned} \tag{A.8}
$$

and similarly,

$$
\operatorname{Tr}\!\left[ \sum_{k=1}^{\chi} B_k^\dagger B_k \right] = \operatorname{Tr}\!\left[ \sum_{k=1}^{\chi} A_k A_k^\dagger \right] = \operatorname{Tr}\!\left[ \sum_{k=1}^{\chi} B_k B_k^\dagger \right] = \operatorname{Tr}[D]. \tag{A.9}
$$

Now we can rewrite Eq. (A.3) based on $V^\dagger V = \mathbb{1} \otimes \mathbb{1}$ as

$$\sum_{k,k'=1}^{\chi} B_k^\dagger B_{k'} \otimes A_k^\dagger A_{k'} = \sum_{k=1}^{\chi} \frac{D_{k,k}}{d^2} B_k^\dagger B_k \otimes \sum_{k=1}^{\chi} \frac{D_{k,k}}{d^2} A_k^\dagger A_k = \mathbb{1} \otimes \mathbb{1} \,. \tag{A.10}$$

This can be viewed as a rank one operator singular value decomposition of the identity. We could have just as well defined local tensors $\tilde{B}_k = \sqrt{D_{k,k}} B_k, \tilde{A}_k = A_k / \sqrt{D_{k,k}}$, which also has a rank one singular value decomposition

$$\sum_{k,k'=1}^{\chi} \tilde{B}_k^\dagger \tilde{B}_{k'} \otimes \tilde{A}_k^\dagger \tilde{A}_{k'} = \sum_{k=1}^{\chi} \frac{D_{k,k}^2}{d^4} B_k^\dagger B_k \otimes \sum_{k=1}^{\chi} A_k^\dagger A_k = \frac{1}{c} \mathbb{1} \otimes c\mathbb{1} \,, \tag{A.11}$$

where the constant $c = \text{Tr}[D]/d^2$ is determined by Eq. (A.8). This implies $\sum_{k=1}^{\chi} A_k^\dagger A_k = c\mathbb{1}$. A similar result is found when weighting the local tensors the other way $\tilde{B}_k = B_k / \sqrt{D_{k,k}}$, $\tilde{A}_k = A_k \sqrt{D_{k,k}}$ implying $\sum_{k=1}^{\chi} B_k^\dagger B_k = c\mathbb{1}$. Using the same argument but with $VV^\dagger = \mathbb{1} \otimes \mathbb{1}$ we find

$$\sum_{k=1}^{\chi} A_k A_k^\dagger = \sum_{k=1}^{\chi} B_k B_k^\dagger = \sum_{k=1}^{\chi} A_k^\dagger A_k = \sum_{k=1}^{\chi} B_k^\dagger B_k = c\mathbb{1} \,. \tag{A.12}$$

Returning to Eq. (A.1), for local unitary QCA, the relevant terms are

$$\begin{aligned}
\text{Tr}\big[(M_L^\dagger M_L)^m\big] &= \text{Tr}\bigg[\big(\sum_k A_k^\dagger A_k\big)^m\bigg] \text{Tr}\bigg[\big(\sum_k B_k B_k^\dagger\big)^m\bigg] \\
&= \text{Tr}[(c\mathbb{1})^m] \text{Tr}[(c\mathbb{1})^m] \\
&= d^4 (\text{Tr}[D]/d^2)^{2m} \\
&= \text{Tr}\big[(M_R^\dagger M_R)^m\big].
\end{aligned} \tag{A.13}$$

This fact assures that the singular values of $M_L$ and $M_R$ are equal. The argument follows from the trace Cayley Hamilton theorem [42] which states that for a complex $n \times n$ matrix $A$, the characteristic function can be written as

$$\begin{aligned}
\chi_A(\lambda) &= \det(\lambda\mathbb{1} - A) \\
&= \lambda^n + P_1 \lambda^{n-1} + P_2 \lambda^{n-2} + \cdots + P_{n-1}\lambda + P_n \,,
\end{aligned} \tag{A.14}$$

where

$$P_k = \sum_{(p_1, p_2, \dots, p_k) \in S_k} \prod_{m=1}^{k} \frac{1}{p_m!} \Big(\frac{-\text{Tr}[A^m]}{m}\Big)^{p_m} \,, \tag{A.15}$$

and $S_k = \{(p_1, p_2, \dots, p_k)\}$ is the set of non-negative integer solutions to the equation $\sum_{r=1}^{k} r p_r = k$. When $\text{Tr}\big[(M_L^\dagger M_L)^k\big] = \text{Tr}\big[(M_R^\dagger M_R)^k\big] \ \forall \ k \in \mathbb{N}$, then the singular values of $M_L^\dagger M_L$ and $M_R^\dagger M_R$ must be equal. These singular values are the squares of the singular values of $M_L$, and $M_R$, which themselves are non negative, hence the singular values of $M_L$ and $M_R$ are equal.

# B Reformulation of the current in terms of the Choi-Jamiolkowski state

It is shown that the information current can be rewritten in terms of the difference in Rényi-2 entropies of the inner product of the Choi-Jamiolkowski state (CJS) associated with $M_L^\dagger M_L$ and

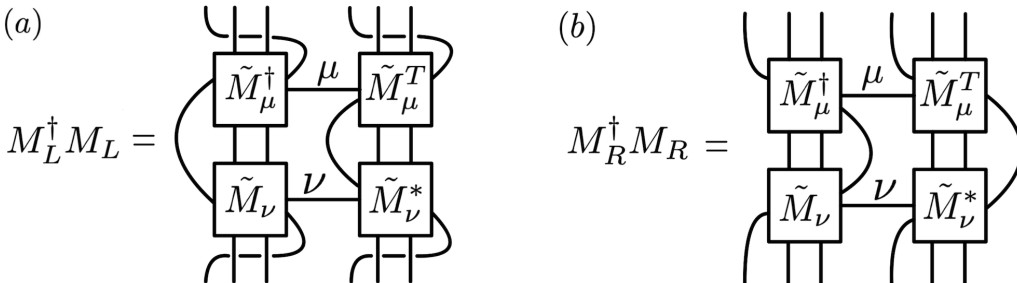

Figure 20: Tensor network description of (a) $M_L^\dagger M_L$, and (b) $M_R^\dagger M_R$, both rewritten in terms of the non-vectorized operators $\tilde{M}_\beta$; $M_\beta^\dagger M_\beta = \left(\sum_\mu \tilde{M}_{\beta_\mu}^\dagger \otimes \tilde{M}_{\beta_\mu}^T\right)\left(\sum_\nu \tilde{M}_{\beta_\nu} \otimes \tilde{M}_{\beta_\nu}^*\right)$. The subscript $\beta \in \{L, R\}$ in the tensor labels is omitted for clarity, as the associated left or right partitioning is given by the regrouping of virtual indices as defined in Fig. 3.

$M_R^\dagger M_R$, respectively. We introduce $\{\tilde{M}_{\beta_\mu}\}_\mu$, $\beta \in \{L, R\}$, as the set of (left or right partitioned) non-vectorized local tensor of the MPO corresponding to $M_\beta = \sum_\mu \tilde{M}_{\beta_\mu} \otimes \tilde{M}_{\beta_\mu}^*$, see Fig. 20. (Note that repeated indices are summed over, here and in subsequent figures in this section.)

Given the map

$$\varepsilon_\beta(\rho) = \sum_{\mu, \nu} \left(\tilde{M}_{\beta_\mu}^\dagger \tilde{M}_{\beta_\nu}\right) \rho \left(\tilde{M}_{\beta_\nu}^\dagger \tilde{M}_{\beta_\mu}\right), \tag{B.1}$$

the CJS is defined by

$$\rho_\beta = \left(\mathbb{1}_{1,2,3} \otimes \varepsilon_{\beta_{4,5,6}}\right)\left(\rho_{|\Phi^+\rangle}\right), \tag{B.2}$$

acting on

$$\rho_{|\Phi^+\rangle} = \left|\Phi_d^+\right\rangle\!\left\langle\Phi_d^+\right|_{1,6} \otimes \left|\Phi_d^+\right\rangle\!\left\langle\Phi_d^+\right|_{2,5} \otimes \left|\Phi_D^+\right\rangle\!\left\langle\Phi_D^+\right|_{3,4}, \tag{B.3}$$

where $\left|\Phi_d^+\right\rangle = \frac{1}{\sqrt{d}}\sum_{j=0}^{d-1}|j\rangle|j\rangle$ are maximally entangled quantum states of two $d$-dimensional qudits; see Fig. 21. $\left|\Phi_2^+\right\rangle$ would represent a Bell pair of two qubits in case of $d = 2$. Considering a QCA with a two-cell neighborhood, the two qudits on sites 5 and 6 represent the physical input states of the QCA on which the MPO, or here the map $\varepsilon$, acts on. They are $d$-dimensional and are each maximally entangled with the qudit at site 1 or 2, respectively. Since the local tensors of the MPO have a virtual degree of freedom of bond dimension $D$, see Fig. 3, we have introduced the $D$-dimensional qudit pair $\left|\Phi_D^+\right\rangle\!\left\langle\Phi_D^+\right|_{3,4}$ on which the virtual degree of freedom of $\varepsilon_\beta$ acts on.

Using the Rényi-2 entropy $S_2(\rho) = -\log \text{Tr}\left[\rho^2\right]$ and the identity $\text{Tr}\left[\rho^2\right] = \langle\rho|\rho\rangle$, where $|\rho\rangle$ is the vectorized state of $\rho$, see Fig. 22, the current in Eq. (5) can then be rewritten in terms of the inner product of the CJS in Eq. (B.2):

$$I = \frac{1}{2}\log\left(\frac{\text{Tr}\left[\rho_L^2\right]}{\text{Tr}\left[\rho_R^2\right]}\right) = \frac{1}{2}\log\left(\frac{\langle\rho_L|\rho_L\rangle}{\langle\rho_R|\rho_R\rangle}\right), \tag{B.4}$$

as shown in Fig. 23. State tomography of $\left|\rho_\beta\right\rangle$ could then determine the current.

## C Proof of the invariance of $I$ under local unitary conjugation of the gates $V$ and $W$

It is derived that $I$ is invariant under conjugation of the local gates $V$ and $W$ with (the same) one-site unitaries $U$; writing $I(V, W) = I\left((U^\dagger \otimes U^\dagger)V(U \otimes U), (U^\dagger \otimes U^\dagger)W(U \otimes U)\right)$.

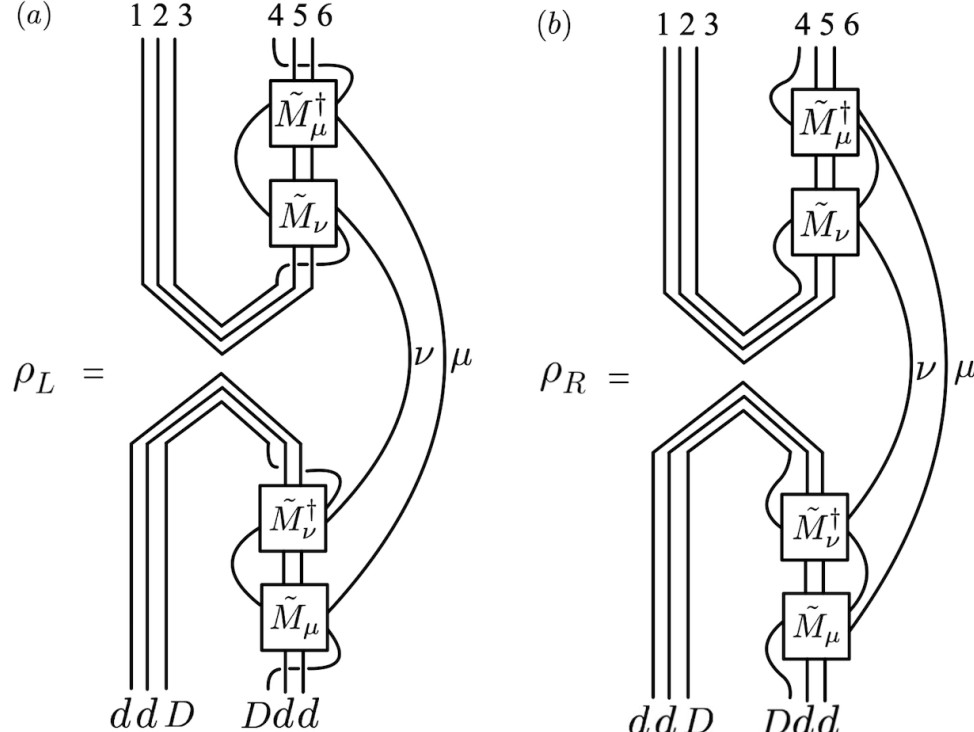

Figure 21: Tensor network description of the CJS associated with the quantum channels (a) $\varepsilon_L(\rho)$ and (b) $\varepsilon_R(\rho)$ shown in Eq. (B.1). $d$ is the physical dimension of each input state, while $D$ represents the bond dimension of the local MPO tensors $\tilde{M}_\mu$.

The invariance follows straight forward from the definition of a unitary matrix, $UU^\dagger = \mathbb{1}$, and the cyclic property of the trace:

$$\mathrm{Tr}\left[\left(M_L^\dagger M_L\right)^2\right] = \mathrm{Tr}\sum_{a,b,c,d=1}^{\chi} (U\otimes U)^\dagger \left(A_a^\dagger \otimes B_b^\dagger\right) (U\otimes U)(U\otimes U)^\dagger\, W^\dagger W\, (U\otimes U)$$

$$\times (U\otimes U)^\dagger (A_a\otimes B_c)\,(U\otimes U)(U\otimes U)^\dagger \left(A_d^\dagger \otimes B_c^\dagger\right)(U\otimes U)$$

$$\times (U\otimes U)^\dagger\, W^\dagger W\,(U\otimes U)(U\otimes U)^\dagger (A_d\otimes B_b)\,(U\otimes U)$$

$$= \mathrm{Tr}\sum_{a,b,c,d=1}^{\chi} \left(A_a^\dagger \otimes B_b^\dagger\right) W^\dagger W\, (A_a\otimes B_c)\left(A_d^\dagger \otimes B_c^\dagger\right) W^\dagger W\,(A_d\otimes B_b), \qquad \text{(C.1)}$$

and analogously for $\mathrm{Tr}\left[\left(M_R^\dagger M_R\right)^2\right]$.

## D Derivation of the current for the reset-swap QCA

The current of the reset-swap QCA is derived as a function of the reset gate $R$ and the set of Pauli operators $\{\sigma^a\}_{a=0}^3$. It is further shown that the current for this map does not change if one would add a local unitary $U$; i.e. $I$ is invariant under "transformations" $R \to RU$, $R \to UR$, or $R \to U^\dagger RU$.

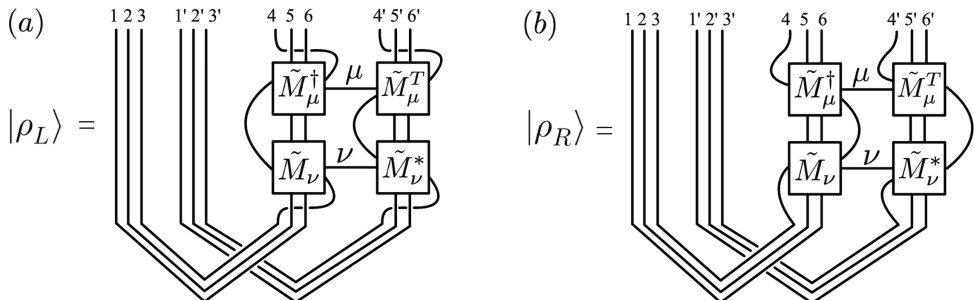

Figure 22: Vectorized version of the CJS of (a) $M_L^\dagger M_L$ and (b) $M_R^\dagger M_R$.

First, the traces of the second moments of $\left(M_{L,R}^\dagger M_{L,R}\right)$ are simplified as illustrated in Fig. 24: the left-hand sites in subfigures (a) and (b) are

$$\mathrm{Tr}\left[\left(M_L^\dagger M_L\right)^2\right] = \frac{1}{2^8}\mathrm{Tr}\Big[\sum_{a,b,c,d,e,f,g,h=0}^3 R_2^\dagger(\sigma^a\otimes\sigma^b)_1\otimes(\sigma^c\otimes\sigma^d)_2 R_1^\dagger$$
$$\times R_1(\sigma^a\otimes\sigma^b)_1\otimes(\sigma^e\otimes\sigma^f)_2 R_2 R_2^\dagger(\sigma^h\otimes\sigma^h)_1\otimes(\sigma^e\otimes\sigma^f)_2 R_1^\dagger$$
$$\times R_1(\sigma^g\otimes\sigma^h)_1\otimes(\sigma^c\otimes\sigma^d)_2 R_2\Big], \tag{D.1a}$$

$$\mathrm{Tr}\left[\left(M_R^\dagger M_R\right)^2\right] = \frac{1}{2^8}\mathrm{Tr}\Big[\sum_{a,b,c,d,e,f,g,h=0}^3 R_2^\dagger(\sigma^a\otimes\sigma^b)_1\otimes(\sigma^c\otimes\sigma^d)_2 R_1^\dagger$$
$$\times R_1(\sigma^g\otimes\sigma^h)_1\otimes(\sigma^c\otimes\sigma^d)_2 R_2 R_2^\dagger(\sigma^g\otimes\sigma^h)_1\otimes(\sigma^e\otimes\sigma^f)_2 R_1^\dagger$$
$$\times R_1(\sigma^a\otimes\sigma^b)_1\otimes(\sigma^e\otimes\sigma^f)_2 R_2\Big], \tag{D.1b}$$

respectively. Separating the operators acting on both, site 1 and site 2, leads to the expressions on the right-hand site of Fig. 24:

$$\mathrm{Tr}\left[\left(M_L^\dagger M_L\right)^2\right] = \frac{1}{2^8}\mathrm{Tr}\Big[\sum_{a,b,g,h=0}^3(\sigma^g\otimes\sigma^h)(\sigma^a\otimes\sigma^b)R^\dagger R(\sigma^a\otimes\sigma^b)(\sigma^g\otimes\sigma^h)R^\dagger R\Big]_1$$
$$\times\mathrm{Tr}\Big[\sum_{c,d,e,f=0}^3(\sigma^c\otimes\sigma^d)(\sigma^e\otimes\sigma^f)RR^\dagger(\sigma^e\otimes\sigma^f)(\sigma^c\otimes\sigma^d)RR^\dagger\Big]_2, \tag{D.2a}$$

$$\mathrm{Tr}\left[\left(M_R^\dagger M_R\right)^2\right] = \frac{1}{2^8}\mathrm{Tr}\Big[\sum_{a,b=0}^3(\sigma^a\otimes\sigma^b)(\sigma^a\otimes\sigma^b)R^\dagger R\sum_{g,h=0}^3(\sigma^g\otimes\sigma^h)(\sigma^g\otimes\sigma^h)R^\dagger R\Big]_1$$
$$\times\mathrm{Tr}\Big[\sum_{c,d=0}^3(\sigma^c\otimes\sigma^d)(\sigma^c\otimes\sigma^d)RR^\dagger\sum_{e,f=0}^3(\sigma^e\otimes\sigma^f)(\sigma^e\otimes\sigma^f)RR^\dagger\Big]_2$$
$$= 2^8\,\mathrm{Tr}\big[R^\dagger RR^\dagger R\big]_1\,\mathrm{Tr}\big[RR^\dagger RR^\dagger\big]_2$$
$$= 2^8\left(\mathrm{Tr}\big[R^\dagger RR^\dagger R\big]\right)^2, \tag{D.2b}$$

where the subscripts 1 and 2 indicate the site all operators in the arguments of the respective traces are acting on. The last two simplifications follow from the identities presented in Fig. 13(b) and (c).

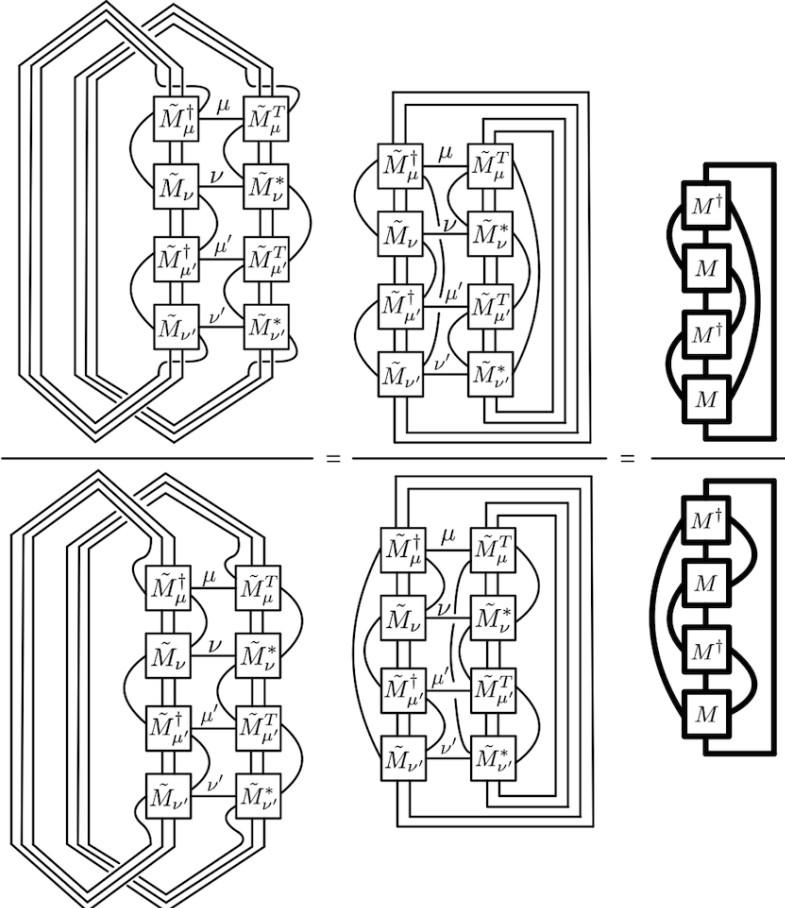

Figure 23: Argument of the current in Fig. 5 rewritten in terms of the inner products of the CJS of $M_L^\dagger M_L$ and $M_R^\dagger M_R$, respectively, shown in Fig. 21 and 22.

The argument of the current is then in total:

$$\frac{\text{Tr}\left[\left(M_L^\dagger M_L\right)^2\right]}{\text{Tr}\left[\left(M_R^\dagger M_R\right)^2\right]} = \frac{1}{(\text{Tr}[R^\dagger RR^\dagger R])^2} \text{Tr}\left[ \sum_{a,b,g,h=0}^{3} (\sigma^g \otimes \sigma^h)(\sigma^a \otimes \sigma^b)R^\dagger R(\sigma^a \otimes \sigma^b)(\sigma^g \otimes \sigma^h)R^\dagger R\right]$$

$$\times \text{Tr}\left[ \sum_{c,d,e,f=0}^{3} (\sigma^c \otimes \sigma^d)(\sigma^e \otimes \sigma^f)RR^\dagger(\sigma^e \otimes \sigma^f)(\sigma^c \otimes \sigma^d)RR^\dagger\right].$$

(D.3)

This expression does not change under composition with a unitary $U$, where $UU^\dagger = \mathbb{1}$, because the three terms

(i) $\text{Tr}\left[R^\dagger RR^\dagger R\right]$,

(ii) $\displaystyle\sum_{a,b=0}^{3} (\sigma^a \otimes \sigma^b)R^\dagger R(\sigma^a \otimes \sigma^b)$,

(iii) $\displaystyle\sum_{a,b=0}^{3} (\sigma^a \otimes \sigma^b)RR^\dagger(\sigma^a \otimes \sigma^b)$,

are invariant under the substitutions $R \to RU$, $R \to UR$, and $R \to URU$. The proof is shown below, where for each substitution the invariance of all three terms in (i) to (iii) is listed.

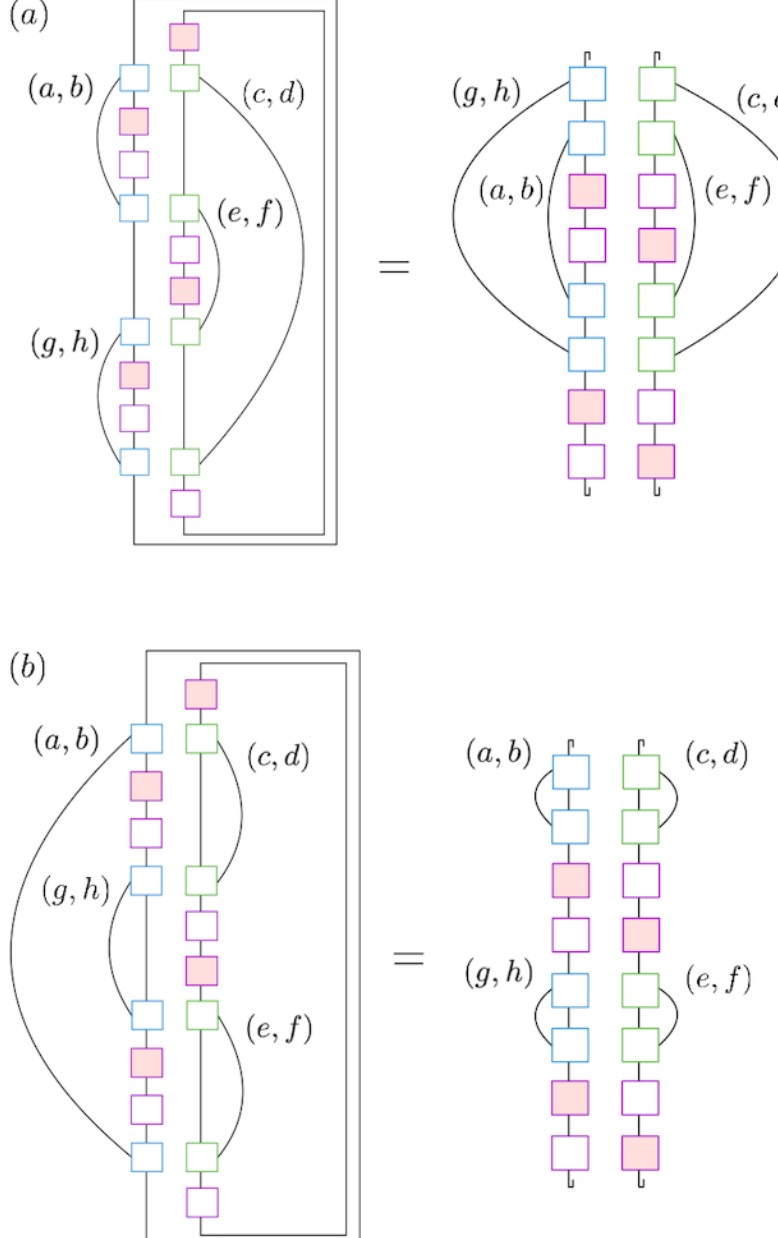

Figure 24: Tensor network description of (a) $\mathrm{Tr}\!\left[\left(M_L^\dagger M_L\right)^2\right]$, (b) $\mathrm{Tr}\!\left[\left(M_R^\dagger M_R\right)^2\right]$, with added summation indices $(a,b),(c,d),(e,f),(g,h)$ used for the derivation of the current of the reset-swap QCA; see Eqs. (D.1) and (D.2).

- $R \to RU$,

$$\text{(i) } \text{Tr}\big[R^\dagger R R^\dagger R\big] = \text{Tr}\big[R R^\dagger R R^\dagger\big] \to \text{Tr}\big[(RU)(U^\dagger R^\dagger)(RU)(U^\dagger R^\dagger)\big] = \text{Tr}\big[R^\dagger R R^\dagger R\big],$$

$$\text{(ii) } \sum_{a,b=0}^{3} (\sigma^a \otimes \sigma^b) R^\dagger R (\sigma^a \otimes \sigma^b) \to \sum_{a,b=0}^{3} (\sigma^a \otimes \sigma^b)(U^\dagger R^\dagger)(RU)(\sigma^a \otimes \sigma^b)$$

$$= \sum_{a,b=0}^{3} (\sigma^a \otimes \sigma^b) R^\dagger R (\sigma^a \otimes \sigma^b),$$

$$\text{(iii) } RR^\dagger \to (RU)(U^\dagger R^\dagger) = RR^\dagger. \tag{D.4}$$

- $R \to UR$,

$$\text{(i) } \text{Tr}\big[R^\dagger R R^\dagger R\big] \to \text{Tr}\big[(R^\dagger U^\dagger)(UR)(R^\dagger U^\dagger)(UR)\big] = \text{Tr}\big[R^\dagger R R^\dagger R\big],$$

$$\text{(ii) } R^\dagger R \to (R^\dagger U^\dagger)(UR) = R^\dagger R,$$

$$\text{(iii) } \sum_{a,b=0}^{3} (\sigma^a \otimes \sigma^b) RR^\dagger (\sigma^a \otimes \sigma^b) \to \sum_{a,b=0}^{3} (\sigma^a \otimes \sigma^b)(UR)(R^\dagger U^\dagger)(\sigma^a \otimes \sigma^b)$$

$$= \sum_{a,b=0}^{3} (\sigma^a \otimes \sigma^b) RR^\dagger (\sigma^a \otimes \sigma^b). \tag{D.5}$$

- $R \to URU$,

$$\text{(i) } \text{Tr}\big[R^\dagger R R^\dagger R\big] \to \text{Tr}\big[(U^\dagger R^\dagger U^\dagger)(URU)(U^\dagger R^\dagger U^\dagger)(URU)\big] = \text{Tr}\big[R^\dagger R R^\dagger R\big],$$

$$\text{(ii) } \sum_{a,b=0}^{3} (\sigma^a \otimes \sigma^b) R^\dagger R (\sigma^a \otimes \sigma^b) \to \sum_{a,b=0}^{3} (\sigma^a \otimes \sigma^b)(U^\dagger R^\dagger U^\dagger)(URU)(\sigma^a \otimes \sigma^b)$$

$$= \sum_{a,b=0}^{3} (\sigma^a \otimes \sigma^b) U^\dagger (R^\dagger R) U (\sigma^a \otimes \sigma^b)$$

$$= \sum_{a,b=0}^{3} (\sigma^a \otimes \sigma^b) R^\dagger R (\sigma^a \otimes \sigma^b),$$

$$\text{(iii) } \sum_{a,b=0}^{3} (\sigma^a \otimes \sigma^b) RR^\dagger (\sigma^a \otimes \sigma^b) \to \sum_{a,b=0}^{3} (\sigma^a \otimes \sigma^b)(U^\dagger R^\dagger U^\dagger)(URU)(\sigma^a \otimes \sigma^b)$$

$$= \sum_{a,b=0}^{3} (\sigma^a \otimes \sigma^b) U^\dagger (R^\dagger R) U (\sigma^a \otimes \sigma^b)$$

$$= \sum_{a,b=0}^{3} (\sigma^a \otimes \sigma^b) R^\dagger R (\sigma^a \otimes \sigma^b). \tag{D.6}$$

Note that in Eqs. (i) the unitary condition is used, $U^\dagger U = \mathbb{1} = UU^\dagger$, while in Eqs. (ii) the unitary $U$ does not change the result, because it acts as a change of basis on the constituent evolution operators $\{\sigma^a\}_{a=0}^{3}$ of the swap operators.

The current is therefore invariant under composition with a unitary finite-depth circuit, and independent of the ordering of the composition with the reset gate.

Nonetheless, an additional local unitary gate would change the current if the QCA is coarse-grained and composed of two or more single time steps. The proof is captured by below

arguments of the current under composition in Eqs. (D.7) to (D.9), and pictured in Fig. 25.

$$\text{Tr}\Big[\big((M^n)^{\otimes^n}\big)_L^\dagger\big((M^n)^{\otimes^n}\big)_L\Big] = \text{Tr}\Big[\sum_{a,b,g,h=0}^{3}(\sigma^g\otimes\sigma^h)(\sigma^a\otimes\sigma^b)\big(U^\dagger R^\dagger\big)^{2n-1}$$
$$\times (RU)^{2n-1}(\sigma^a\otimes\sigma^b)(\sigma^g\otimes\sigma^h)\big(U^\dagger R^\dagger\big)^{2n-1}(RU)^{2n-1}\Big]_1$$
$$\times \text{Tr}\Big[\sum_{c,d,e,f=0}^{3}(\sigma^c\otimes\sigma^d)(\sigma^e\otimes\sigma^f)(RU)^{2n-1}$$
$$\times \big(U^\dagger R^\dagger\big)^{2n-1}(\sigma^e\otimes\sigma^f)(\sigma^c\otimes\sigma^d)(RU)^{2n-1}\big(U^\dagger R^\dagger\big)^{2n-1}\Big]_2$$
$$\times \text{Tr}\Big[\big((M^{n-1})^{\otimes^{n-1}}\big)_L^\dagger\big((M^{n-1})^{\otimes^{n-1}}\big)_L\Big], \tag{D.7}$$

$$\text{Tr}\Big[\big((M^n)^{\otimes^n}\big)_R^\dagger\big((M^n)^{\otimes^n}\big)_R\Big] = \Big(\text{Tr}\big[(U^\dagger R^\dagger)^{2n-1}(RU)^{2n-1}\big]\Big)^2\text{Tr}\Big[\big((M^{n-1})^{\otimes^{n-1}}\big)_R^\dagger\big((M^{n-1})^{\otimes^{n-1}}\big)_R\Big], \tag{D.8}$$

$$\frac{\text{Tr}\Big[\big((M^n)^{\otimes^n}\big)_L^\dagger\big((M^n)^{\otimes^n}\big)_L\Big]}{\text{Tr}\Big[\big((M^n)^{\otimes^n}\big)_R^\dagger\big((M^n)^{\otimes^n}\big)_R\Big]} = \frac{1}{(\text{Tr}\big[(U^\dagger R^\dagger)^{2n-1}(RU)^{2n-1}\big])^2}$$
$$\times \text{Tr}\Big[\sum_{a,b,g,h=0}^{3}(\sigma^g\otimes\sigma^h)(\sigma^a\otimes\sigma^b)\big(U^\dagger R^\dagger\big)^{2n-1}(RU)^{2n-1}$$
$$\times (\sigma^a\otimes\sigma^b)(\sigma^g\otimes\sigma^h)\big(U^\dagger R^\dagger\big)^{2n-1}(RU)^{2n-1}\Big]_1$$
$$\times \text{Tr}\Big[\sum_{c,d,e,f=0}^{3}(\sigma^c\otimes\sigma^d)(\sigma^e\otimes\sigma^f)(RU)^{2n-1}(U^\dagger R^\dagger)^{2n-1}$$
$$\times (\sigma^e\otimes\sigma^f)(\sigma^c\otimes\sigma^d)(RU)^{2n-1}\big(U^\dagger R^\dagger\big)^{2n-1}\Big]_2$$
$$\times \frac{\text{Tr}\Big[\big((M^{n-1})^{\otimes^{n-1}}\big)_L^\dagger\big((M^{n-1})^{\otimes^{n-1}}\big)_L\Big]}{\text{Tr}\Big[\big((M^{n-1})^{\otimes^{n-1}}\big)_R^\dagger\big((M^{n-1})^{\otimes^{n-1}}\big)_R\Big]}. \tag{D.9}$$

# E  Derivation of $W^\dagger W$ for the directed amplitude damping map

It is shown that for the directed amplitude damping channel discussed in Sec. 3.4, $W^\dagger W$ is not separable when $p > 0$.

The map is defined by the Kraus operators

$$K_0 = \begin{pmatrix} 1 & 0 & 0 & 0 \\ 0 & 1 & 0 & 0 \\ 0 & 0 & \sqrt{1-p} & 0 \\ 0 & 0 & 0 & 1 \end{pmatrix}, \qquad K_1 = \begin{pmatrix} 0 & 0 & \sqrt{p} & 0 \\ 0 & 0 & 0 & 0 \\ 0 & 0 & 0 & 0 \\ 0 & 0 & 0 & 0 \end{pmatrix}, \tag{E.1}$$

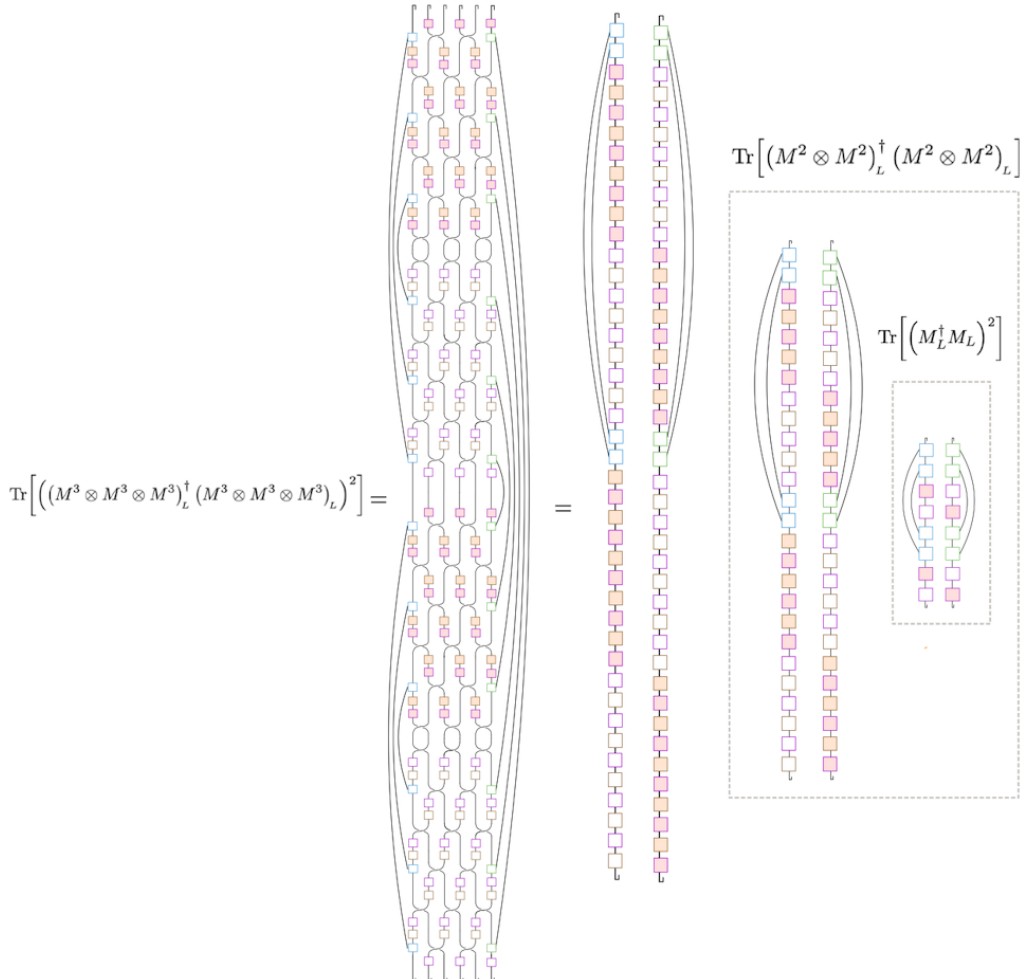

Figure 25: Tensor network description of $\mathrm{Tr}\left[\left((M^n)^{\otimes^n}\right)_L^{\dagger}\left((M^n)^{\otimes^n}\right)_L\right]$ in Eq. (D.9) for the unitary-reset-swap QCA ($\mathrm{SWAP}_{1,2}\,R_1\,U_1$) under composition of $n=3$ time steps. The same notation as is Figs. 12, and 13 is used, where the brown tensors represent the unitary superoperator, $(U \otimes U^*)$.

which determine the vectorized transfer matrix:

$$
\begin{aligned}
W &= K_0 \otimes K_0 + K_1 \otimes K_1 \\
&= (|0\rangle\langle 0| \otimes |0\rangle\langle 0| + |0\rangle\langle 0| \otimes |1\rangle\langle 1| + \sqrt{1-p}\,|1\rangle\langle 1| \otimes |0\rangle\langle 0| + |1\rangle\langle 1| \otimes |1\rangle\langle 1|) \\
&\quad \otimes (|0\rangle\langle 0| \otimes |0\rangle\langle 0| + |0\rangle\langle 0| \otimes |1\rangle\langle 1| + \sqrt{1-p}\,|1\rangle\langle 1| \otimes |0\rangle\langle 0| + |1\rangle\langle 1| \otimes |1\rangle\langle 1|) \\
&\quad + p\,(|0\rangle\langle 1| \otimes |0\rangle\langle 0|) \otimes (|0\rangle\langle 1| \otimes |0\rangle\langle 0|).
\end{aligned}
\tag{E.2}
$$

Applying the basis-change transformation from Eq. (11),

$$
W \to (\mathbb{1} \otimes \hat{\Sigma} \otimes \mathbb{1})\, W\, (\mathbb{1} \otimes \hat{\Sigma} \otimes \mathbb{1}),
\tag{E.3}
$$

rearranges the order of subsystems in the tensor product, such that the operators can be com-

bined which act on the same physical site, indicated by subscripts 1 and 2:

$$
\begin{aligned}
W = & |00\rangle\langle00|_1 \otimes \mathbb{1}_2 + |01\rangle\langle01|_1 \otimes (\sqrt{1-p}\,|00\rangle\langle00| \\
& + |01\rangle\langle01| + \sqrt{1-p}\,|10\rangle\langle10| + |11\rangle\langle11|)_2 \\
& + |10\rangle\langle10|_1 \otimes (\sqrt{1-p}\,|00\rangle\langle00| + \sqrt{1-p}\,|01\rangle\langle01| \\
& + |10\rangle\langle10| + |11\rangle\langle11|)_2 + |11\rangle\langle11|_1 \otimes ((1-p)\,|00\rangle\langle00| \\
& + \sqrt{1-p}\,|01\rangle\langle01| + \sqrt{1-p}\,|10\rangle\langle10| + |11\rangle\langle11|)_2 \\
& + p\,|00\rangle\langle11|_1 \otimes |00\rangle\langle00|_2\,.
\end{aligned}
\tag{E.4}
$$

Now one can write

$$
W^\dagger W = \sum_{r,s=1}^{16} c_{r,s} P(r)_1 \otimes P(s)_2\,,
\tag{E.5}
$$

where $\{P(a)\}_{a=1}^{16}$ is an orthonormal basis for two qubits. By explicit calculation one finds that for $p > 0$, the matrix $c$ has four singular values whereas for $p = 0$ there is only one as expected for the unitary case. Hence $W^\dagger W$ is not separable for $p > 0$.

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
