# Peer review of "Information flow in non-unitary quantum cellular automata"

_SciPost Physics, doi:SciPost Phys. 16, 014 (2024)_

## Round 1 · Referee Report · Anonymous (Referee 1) · 2023-11-7

Report

The manuscript deals with an important question of quantum dynamics, namely how to measure information flow in a quantum system.

In Quantum Cellular Automata, they have an index theory to quantify information flow over a boundary. This quantity, the GNVW (Gross, Nesme, Vogts, Werner) index, has many advantageous properties. E.g., it is invariant under finite depth local circuits.

The paper presents the index theory in the matrix product unitary (MPU) framework, where the index is given with the ranks of the MR and ML matrices. It is stressed that the index is the same for all locally equivalent unitary QCA.

However, it is not defined for non-unitary dynamics. Thus, the authors propose a new measure that is called information current. They use matrix product operators (MPO) for the description of the systems. They define it he information flow as the function of the singular values of MR and ML. At the end, they present a formula with the Rényi entropy of those operators and denote the quantity by I.

They list a number if important properties of I:

  • It is locally computable.
  • I=0 for unitary finite depth circuits
  • I is not invariant under blocking, which is expected due to its nonlinear nature.
  • I is not additive under decomposition.
  • I=0 if the QCA is swap symmetric, which can be interpreted as not having a preferred direction for the information flow.

The paper presents several very relevant examples.

I find the manuscript very well written, with an excellent introduction. The main result, the definition of the information current, is very important. It presented very clearly. It is supported with a detailed study of the newly defined quantity. Thus, I suggest the publication of the manuscript in SciPost Physics.

Requested changes

Comments:

Maybe, I would suggest to add space in the following cases:

“Sec.II” --> “Sec. II”

and all similar occurrences of “Sec.” without an additional space. Similarly, I suggest

“Fig.2(b-d)” --> “Fig. 2(b-d)”

I would also suggest to following change

“Sec.’s IIIB and III C” --> “Sections IIIB and III C”

---

## Round 1 · Referee Report · Anonymous (Referee 2) · 2023-11-22

Report

The authors study information flow in non-unitary Quantum Cellular Automata (QCA). Unitary QCA are a well-studied class of discrete unitary dynamics which preserve locality of operators. They are characterized by an index which quantifies the flow of information. This index is topological, in the sense that it is robust against continuous deformation. In this work, the authors aim at extending the notion of the index in the case the system is not isolated and the evolution is not unitary.

The main result of the paper is a candidate for a "non-unitary" index, which has a number of nice properties. In particular, it coincides with the conventional index for the unitary case, and gives intuitive results in a few concrete examples. Although this index is not topological, the authors argue that this should be expected, and that the defined index can nevertheless be practically useful.

The question of generalizing the theory of unitary QCA to the non-unitary case is very natural (for instance, experimentally we never have an ideally isolated situation), but technically very hard. This is why relatively few works have touched upon the subject in the past and why I think the results obtained in this paper are very important.

I think the authors make a very good job in review the existing literature. The paper is also extremely clear and I appreciated how it discusses all the features of the non-unitary index, including its potential limitations.

Finally, I think the topic is timely and of interest for a broad audience.

For the reasons above, I recommend publication of the draft, essentially as is.

---

## Editorial Decision

published